# The UFM1 system regulates ER-phagy through the ufmylation of CYB5R3

Ryosuke Ishimura[1], Afnan H. El-Gowily [1,2], Daisuke Noshiro [3], Satoko Komatsu-Hirota[1], Yasuko Ono[4], Mayumi Shindo[5], Tomohisa Hatta[6], Manabu Abe [7], Takefumi Uemura[8], Hyeon-Cheol Lee-Okada [9], Tarek M. Mohamed[2], Takehiko Yokomizo[9], Takashi Ueno[10], Kenji Sakimura[7], Tohru Natsume [6], Hiroyuki Sorimachi [4], Toshifumi Inada [11], Satoshi Waguri [8], Nobuo N. Noda [3] & Masaaki Komatsu [1] ✉

Protein modification by ubiquitin-like proteins (UBLs) amplifies limited genome information and regulates diverse cellular processes, including translation, autophagy and antiviral pathways. Ubiquitin-fold modifier 1 (UFM1) is a UBL covalently conjugated with intracellular proteins through ufmylation, a reaction analogous to ubiquitylation. Ufmylation is involved in processes such as endoplasmic reticulum (ER)-associated protein degradation, ribosome-associated protein quality control at the ER and ER-phagy. However, it remains unclear how ufmylation regulates such distinct ER-related functions. Here we identify a UFM1 substrate, NADH-cytochrome b5 reductase 3 (CYB5R3), that localizes on the ER membrane. Ufmylation of CYB5R3 depends on the E3 components UFL1 and UFBP1 on the ER, and converts CYB5R3 into its inactive form. Ufmylated CYB5R3 is recognized by UFBP1 through the UFM1-interacting motif, which plays an important role in the further uyfmylation of CYB5R3. Ufmylated CYB5R3 is degraded in lysosomes, which depends on the autophagy-related protein Atg7- and the autophagy-adaptor protein CDK5RAP3. Mutations of *CYB5R3* and genes involved in the UFM1 system cause hereditary developmental disorders, and ufmylation-defective *Cyb5r3* knock-in mice exhibit microcephaly. Our results indicate that CYB5R3 ufmylation induces ER-phagy, which is indispensable for brain development.

Ubiquitin-fold modifier 1 (UFM1) is a ubiquitin-like protein[1] whose biological functions are the least understood of all known ubiquitin-like proteins (UBLs)[2]. UFM1 is synthesized as a pro-form, and the two C-terminal amino acids of proUFM1 are cleaved by specific proteases, UFSP1 and UFSP2[3–5]. This results in the mature form, which exposes the glycine residue essential for conjugation[3]. Mature UFM1 is then activated via the formation of a high-energy thioester bond with UBA5, a UFM1-specific E1 enzyme[1]. Activated UFM1 is then transferred to the UFM1-specific E2 enzyme, UFC1[1], and covalently bound to the target protein(s) via a UFM1-specific E3 enzyme complex consisting of UFL1,

UFBP1 (also called DDRGK1 or C20orf116), and CDK5RAP3[6–9]. Of these E3 components, UFL1 and UFBP1 are essential for ufmylation[10,11], while CDK5RAP3 seems to be involved in the regulation of E3 activity such as poly-ufmylation[9] and direct ufmylation to the ribosomal protein RPL26[9]. This reaction system is reversible as covalently bound UFM1 can be cleaved by cysteine protease UFSP2[3,12]. None of the genes involved in UFM1 modification exist in fungi, but they are conserved in plants and metazoans[13]. Impairment of the UFM1 system causes defective erythroid differentiation and neurogenesis in mice[7,14–16], and genetic mutations in *UFM1*, *UBA5*, and *UFC1* diminish ufmylation and

lead to hereditary pediatric encephalopathy[16–18]. *UFSP2* mutations also cause the autosomal dominant disorders Beukes hip dysplasia and spondyloepimetaphyseal dysplasia[19,20]. The UFM1 system has been thought to be linked to ER stress and ER-associated protein degradation (ERAD)[13]. However, several lines of evidence have demonstrated important roles for ufmylation not in ERAD, but in ribosome-associated protein quality control RQC at the ER[21,22] and in selective autophagy of the ER (ER-phagy)[23]. The molecular mechanism(s) by which ER homeostasis is regulated by ufmylation remains largely unclear. Herein, we show that ufmylation of the ER membrane-localizing protein CYB5R3 is a signal for ER-phagy, which is indispensable for neuronal development.

## Results

### Identification of CYB5R3 as a substrate of UFM1

UFBP1 contains an N-terminal signal peptide and localizes to the ER[6]. Thus, we tested whether UFBP1 tethered UFL1 to the ER. In agreement with previous reports[6,10,23], an immunoprecipitation assay showed an interaction between endogenous UFL1 and UFBP1 (Supplementary Fig. 1a). We confirmed the direct interaction between UFL1 and UFBP1 by an in vitro pull-down assay (Supplementary Fig. 1b). To investigate whether UFBP1 can translocate UFL1 to ER, we generated *UFBP1* knockout HeLa cells (Supplementary Fig. 2). GFP-tagged UFL1 (GFP-UFL1) was present throughout the cytoplasm in the *UFBP1* knockout HeLa cells (Supplementary Fig. 1c). When we co-transfected GFP-UFL1 together with MYC-tagged UFBP1 (UFBP1-MYC) in the *UFBP1* knockout cells, both GFP-UFL1 and UFBP1-MYC co-localized on KDEL-positive ER (Supplementary Fig. 1c). These results suggest that UFL1 forms a stable complex with UFBP1 on the ER and becomes an ER-resident E3 ligase for UFM1. The co-expression of UFL1 and UFBP1 promoted UFM1 conjugate formation in only the microsomal fraction (Fig. 1a), which was enhanced by loss of *UFSP2* (Fig. 1a). Ufmylation of the ribosomal protein RPL26 was not observed in HEK293T cells despite overexpression of the E3 components UFL1 and UFBP1 (Fig. 1a), but it was prominently detected in *UFSP2* knockout HEK293T cells that do not release conjugated UFM1, as previously reported[21]. Expression of UFL1 and UFBP1 did not affect levels of ufmylated RPL26 (Fig. 1a). To identify UFM1 substrate(s) on the ER, we expressed the following in HEK293T cells: FLAG- and His-tagged mature UFM1 (FLAG-His-UFM1ΔC2) or a conjugation-defective FLAG-His-UFM1ΔC3 as a negative control, together with MYC-tagged UFL1 (MYC-UFL1) and UFBP1 (UFBP1-MYC). We immunoprecipitated the lysates with anti-FLAG antibody, and then pulled down the immunoprecipitants with Ni-NTA agarose under denaturing conditions. The resulting samples were subjected to SDS-PAGE, the bands were excised and digested with trypsin and analyses were performed with a mass spectrometer (Fig. 1b). We summarized proteins that were present only in cells expressing FLAG-His-UFM1ΔC2 (Supplementary Data 1). In an independent experiment, we screened proteins whose interactions with UFM1 were dependent on the expression of both UFL1 and UFBP1 (Supplementary Data 2). We focused on NADH-cytochrome b5 reductase 3 (CYB5R3) since this protein was identified by both screenings and is known to be anchored on the ER and/or mitochondrial membranes by myristoylation of the N-terminal Gly and a membrane-bound domain[24] (Fig. 1c).

To investigate whether CYB5R3 was covalently conjugated with UFM1 in HEK293T cells, we expressed His- and FLAG-tagged CYB5R3 (CYB5R3-His-FLAG) together with MYC-tagged mature UFM1 (MYC-UFM1). Subsequently, we pulled down the lysates with Ni-NTA agarose under denaturing conditions. When the E3 component UFL1 was co-overexpressed, we observed a band representing the MYC-UFM1-CYB5R3-His-FLAG conjugate, which was further enhanced by the expression of UFBP1 (Fig. 1d). In *UFSP2*-deficient HEK293T cells, the conjugation was observed even without expression of the E3 components, and the level was enhanced by UFL1 (Fig. 1d). Next, we investigated which lysine residue of CYB5R3 was modified by UFM1. To do this, we created a series of KR mutants in which each lysine residue was substituted with an arginine residue, expressed the mutants in *UFSP2* knockout HEK293T cells, and conducted a pull-down assay with Ni-NTA agarose under denaturing conditions. As shown in Fig. 1e, only the K214R mutant (CYB5R3^K214R) lost the ability to conjugate with UFM1. The lysine residue was well conserved among species (Fig. 1f). To conduct an in vitro ufmylation assay, we purified recombinant UFM1, UBA5, and UFC1, as well as CYB5R3ΔN26, which lacked a membrane anchor region, from *Escherichia coli*, and FLAG-UFL1 and UFBP1-MYC from *UFC1*-knockout HEK293T cells (Supplementary Fig. 3a, b). When recombinant CYB5R3ΔN26 was mixed with UFM1, UBA5, UFC1, FLAG-UFL1, and UFBP1-MYC, we did not observe any CYB5R3ΔN26 conjugation with UFM1 (Supplementary Fig. 3c). We concluded that the in vitro ufmylation assay was working since under these reaction conditions, UFBP1-MYC was conjugated with UFM1 (Supplementary Fig. 3d). We speculated that this assay lacked one or more factors required for CYB5R3 ufmylation. Indeed, when we used a microsomal fraction prepared from *UFBP1*-deficient HEK293T cells instead of CYB5R3ΔN26, ufmylation of CYB5R3 occurred dependently of E1-, E2-, and E3 (Fig. 1g). The level of the UFM1-CYB5R3 conjugate was increased in *UFSP2*-deficient HEK293T cells (Fig. 1d), suggesting that this conjugation was reversible. Actually, in *UFSP2*-deficient HEK293T cells, overexpression of UFSP2 but not the active-site mutant UFSP2^C302A decreased the amount of the UFM1-CYB5R3 conjugate (Fig. 1h).

### Ufmylation of CYB5R3 occurs on the endoplasmic reticulum

We divided cell homogenates prepared from HEK293T cells into microsomal and cytoplasmic fractions, and immunoprecipitated both fractions with anti-CYB5R3 antibody followed by immunoblot analysis. As expected, endogenous free CYB5R3 was mainly recovered in the microsomal fraction (Fig. 2a). We also observed endogenous ufmylated CYB5R3 only in the microsomal fraction; its levels increased with expression of UFL1 and UFBP1 (Fig. 2a). In *UFSP2* knockout HEK293T cells, the endogenous CYB5R3 conjugated with UFM1 was clearly recognized, and it was significantly decreased by knockdown of *UFL1* and of *UFBP1* (Fig. 2b). To clarify the ufmylation of CYB5R3 on the ER, we generated *CYB5R3 UFSP2* double-knockout HEK293T cells (Supplementary Fig. 2). When we co-transfected these cells with CYB5R3-His-FLAG, GFP-tagged mature UFM1 (GFP-UFM1), MYC-UFL1 and UFBP1-MYC, we clearly detected GFP-UFM1-CYB5R3-His-FLAG conjugate in the microsomal fraction (Fig. 2c). The GFP-UFM1-CYB5R3-His-FLAG conjugate was not present following expression of CYB5R3^K214R-His-FLAG (Fig. 2c). CYB5R3ΔN26, which lacks both myristoylation of the N-terminal glycine and a membrane-bound domain, was fractionated into only the cytoplasmic fraction and was minimally conjugated with UFM1 (Fig. 2c). CYB5R3-His-FLAG detected in the cytoplasmic fraction upon overexpression was considered to be the N-terminal truncated form since it was almost as mobile as CYB5R3ΔN26 (Fig. 2c). Immunofluorescence analysis showed extensive co-localization of C-terminus GFP-tagged CYB5R3 (CYB5R3-GFP) with an ER marker protein, PDI (Fig. 2d). This co-localization was not observed in the case of CYB5R3ΔN26, and the mutant diffused throughout the cytoplasm (Fig. 2d). These results suggest that CYB5R3 ufmylation occurs on the ER.

### Enzymatic activity of CYB5R3 is lost by the ufmylation

CYB5R3 catalyzes the transfer of reducing equivalents from NADH to cytochrome b5, which then acts as an electron donor. CYB5R3 consists of an N-terminal long hydrophobic stretch sufficient to tightly anchor to the ER, and also FAD- and NADH-binding domains (Fig. 1c)[25]. Previous crystallographic analysis of human CYB5R3 revealed that the FAD and NADH domains interact with each other to form one globular fold containing a large groove that binds FAD[26] (Fig. 3a). A different crystal form of human CYB5R3 showed a dissimilar arrangement of

FAD and NADH domains compared to the original structure[26] (Supplementary Fig. 4), indicating the intrinsic flexibility of this protein. High-speed atomic force microscopy (HS-AFM) showed that two globular lobes of CYB5R3, which corresponds to the NADH and FAD domains, sometimes separated. We measured the distance between the two lobes of imaged CYB5R3. All histograms from the data of 6 molecules were well fitted to double-Gaussian distribution whose peaks were at 2.3–2.9 nm (1st peak) and 4.2–5.1 nm (2nd peak). This suggests that CYB5R3 can adopt an open conformation with the two domains distanced from each other (Fig. 3b, Supplementary Fig. 5, and

Supplementary Movie 1). The states in which two lobes are close together are similar to the simulated AFM image of the crystal structure (PDB ID: 1UMK) and the distance between the two lobes of the simulated image was 2.5–3.0 nm. We manually modeled an open conformation of CYB5R3 by changing the arrangement between NADH and FAD domains at the hinge region (Leu147) so that it gives a predicted AFM image similar to the experimental one (Fig. 3b, c). The distance between the two lobes of the simulated open conformation was 4.0–4.7 nm, consistent with the value of the 2nd peak of the histogram obtained from the HS-AFM experiment. Since Lys214, the

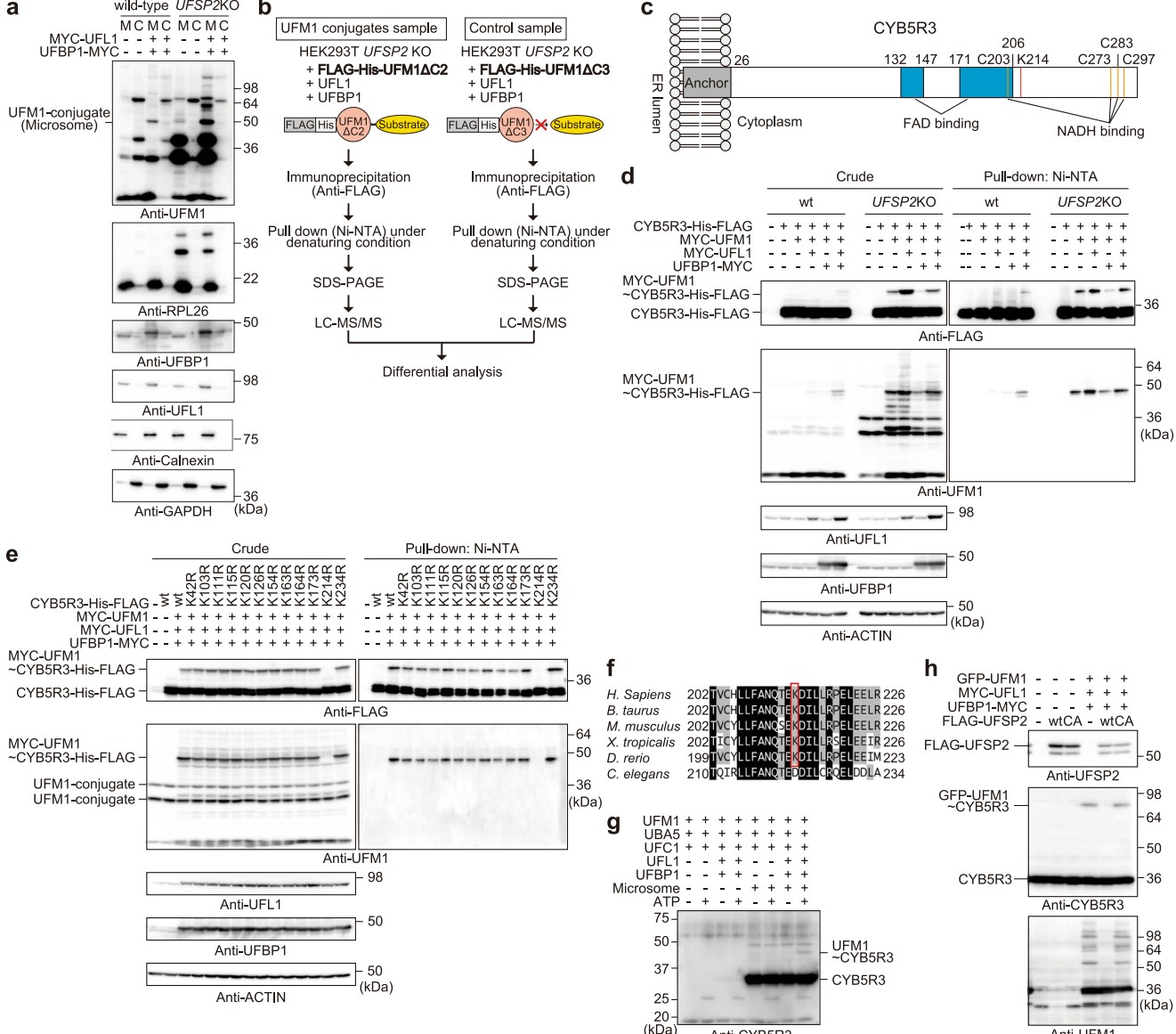

**Fig. 1 | CYB5R3 is a bona fide substrate for UFM1. a** Immunoblot analysis. Wild-type or *UFSP2*[−/−] HEK293T expressing UFL1 and UFBP1 were fractionated into microsomal (M) and cytoplasmic (C) fractions followed by immunoblot analysis. Data shown are representative of three separate experiments. **b** An experimental approach to identify ufmylated proteins. **c** Domain structure of CYB5R3. Membrane anchor, FAD-binding site, and lysine residue for UFM1 conjugation are indicated. **d** Immunoblot analysis. The indicated constructs were transfected into parental or *UFSP2*[−/−] HEK293T. 48 h after the transfection, the cell lysates were pulled down with Ni-NTA agarose under denaturing conditions followed by immunoblot analysis. Data shown are representative of three separate experiments. **e** Immunoblot analysis. A series of CYB5R3KR mutants together with MYC-UFM1, MYC-UFL1, and UFBP-MYC were expressed

in *UFSP2*[−/−] HEK293T. The cell lysates were analyzed as described in **d**. Data shown are representative of three separate experiments. **f** Alignment of the region around K214 of CYB5R3. **g** In vitro conjugation assay. Recombinant UFM1, UBA5, and UFC1 from *E. coli* and FLAG-UFL1 and UFBP1-MYC from *UFC1*-knockout HEK293T cells were incubated with a microsomal fraction prepared from *UFBP1*-deficient HEK293T cells in the presence or absence of ATP for 90 min, and the mixture was subjected to SDS-PAGE followed by immunoblot analysis with CYB5R3 antibody. Data shown are representative of three separate experiments. **h** Immunoblot analysis. The indicated constructs were transfected into *UFSP2*[−/−] HEK293T. 48 h after transfection, cell lysates were subjected to immunoblot analysis. Data shown are representative of three separate experiments. Source data are provided as a Source Data file.

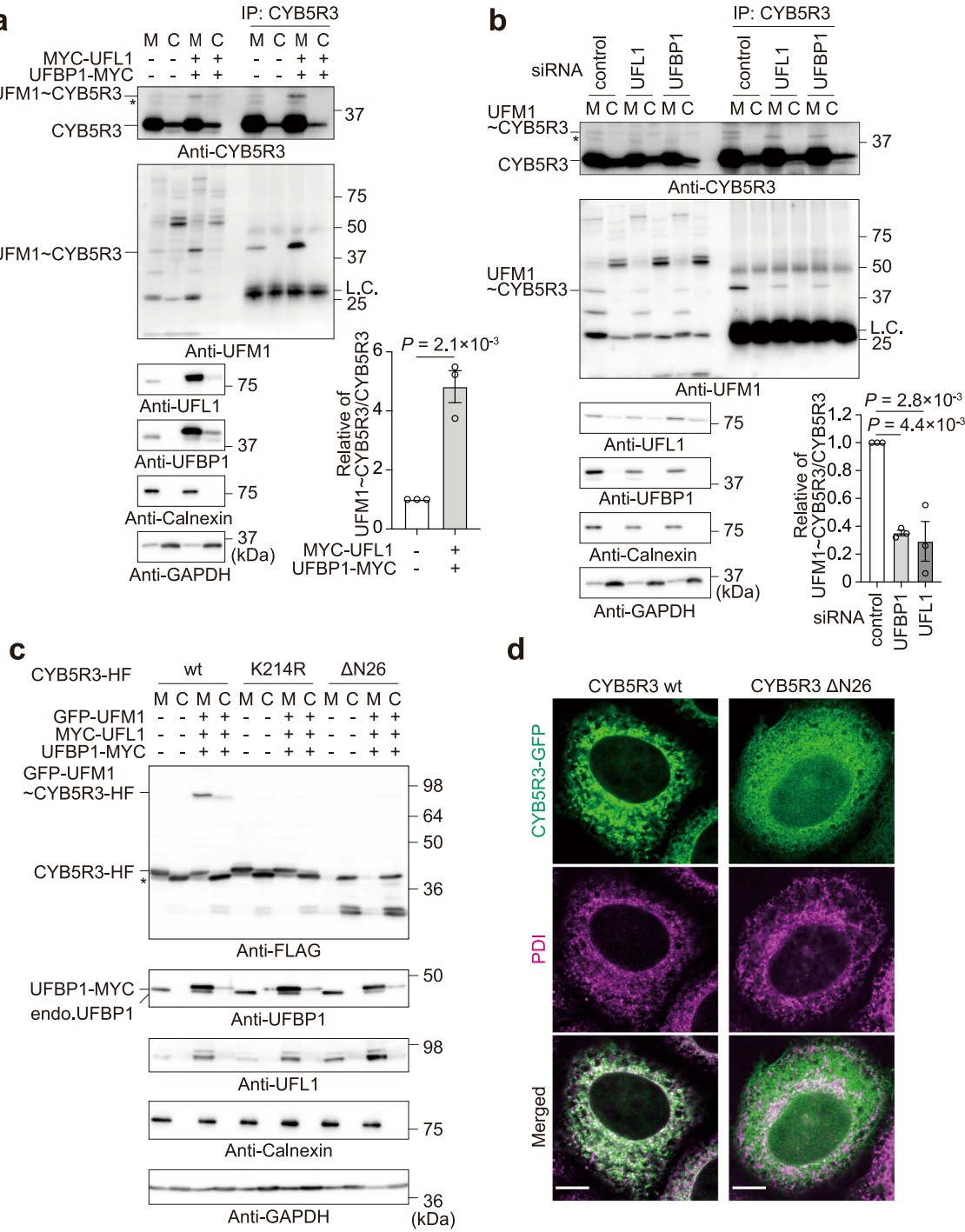

**Fig. 2 | Ufmylation of CYB5R3 on the ER. a** Immunoprecipitation assay. HEK293T expressing UFL1 and UFBP1 were fractionated into microsomal (M) and cytoplasmic (C) fractions. The fractions were denatured with TNE with SDS (final 1%), diluted 10-fold with TNE, and then immunoprecipitated with CYB5R3 antibody followed by immunoblot analyses. Data shown are representative of three separate experiments. Bar graphs show the relative value of the ratio of endogenous ufmylated CYB5R3 to CYB5R3 was 1 when neither overexpression nor knockdown of any E3 components was performed. Data are means ± s.e. Statistical analysis was performed by two-sided Welch's *t* test. **b** Immunoprecipitation assay. *UFL1* or *UFBP1* in *UFSP2*⁻/⁻ HEK293T were knocked down, and the cells were fractionated into microsomal (M) and cytoplasmic (C) fractions. The fractions were analyzed as described in **a**. Data shown are representative of four separate experiments. Bar graphs show the relative value of the ratio of endogenous ufmylated CYB5R3 to CYB5R3 was 1 when neither overexpression nor

knockdown of any E3 components was performed. Data are means ± s.e. Statistical analysis was performed by Šidák's multiple comparison test after one-way ANOVA. **c** Immunoblot analysis. The indicated constructs were transfected into *CYB5R3*⁻/⁻ *UFSP2*⁻/⁻ HEK293T. 48 h after transfection, the cell lysates were fractionated into microsomal (M) and cytoplasmic (C) fractions followed by immunoblot analysis. Data shown are representative of three separate experiments. Asterisk indicates a degradative product of CYB5R3-His-3xFLAG. Since the CYB5R3 in the cytoplasmic fraction (indicated by asterisk) is almost as mobile as ΔN26, it is considered to be the N-terminal truncated form. **d** Immunofluorescence analysis. GFP-tagged wild-type (CYB5R3-GFP) or ER-localization-defective CYB5R3 (CYB5R3ΔN26-GFP) were transfected into *CYB5R3*⁻/⁻ HeLa. 48 h after transfection, the cells were immunostained with anti-PDI antibody. Bars: 10 μm. Data shown are representative of three independent experiments. Source data are provided as a Source Data file.

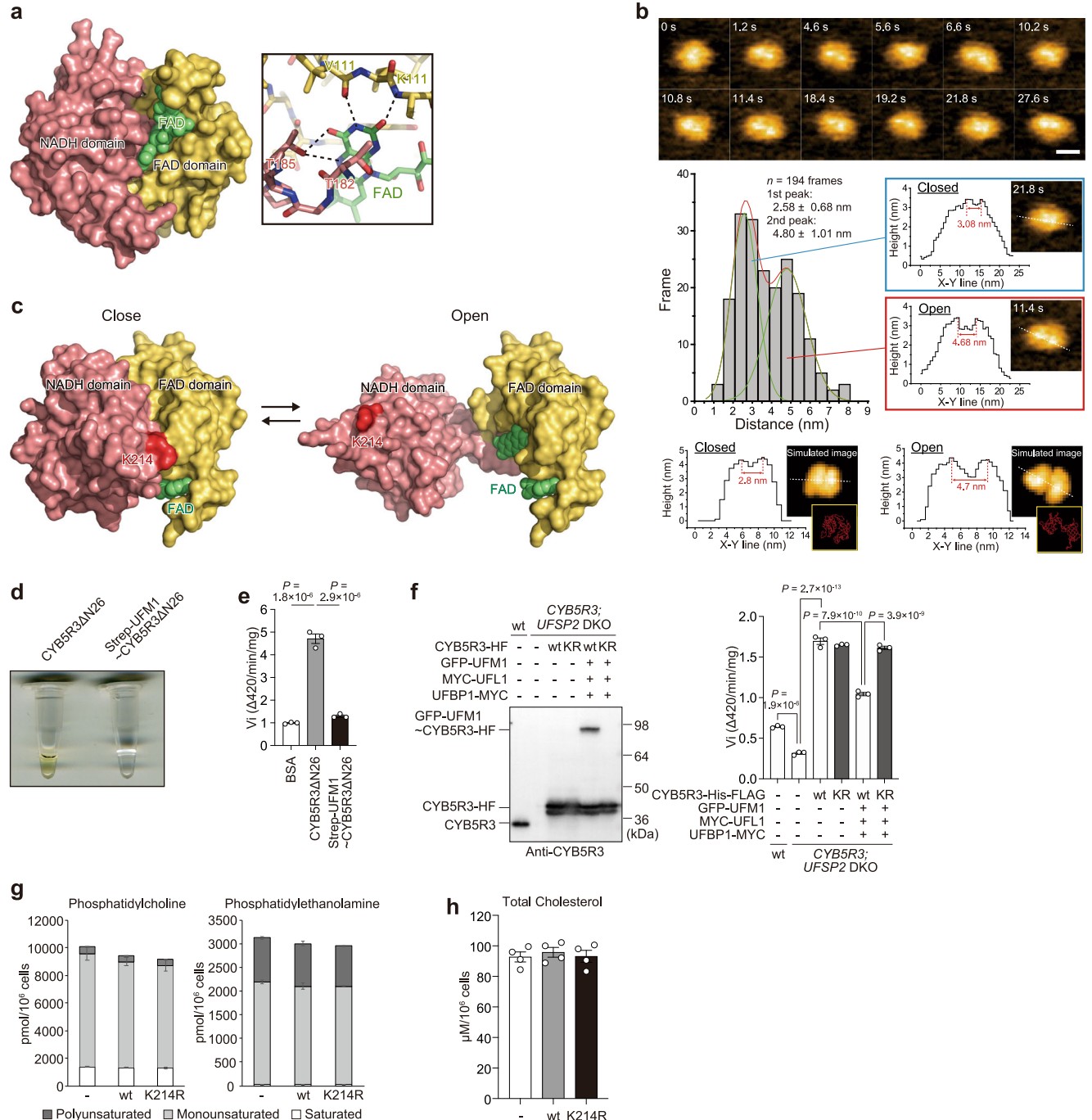

**Fig. 3 | Loss of CYB5R3 enzymatic activity after ufmylation. a** Left, crystal structure of human CYB5R3 (PDB 1UMK). FAD and NADH domains are colored yellow and salmon pink, respectively, whereas bound FAD is colored green. Right, close-up view of the domain interface that recognizes FAD. Broken lines indicate possible hydrogen bonds. **b** Top, Successive HS-AFM images of CYB5R3. Scale, 10 nm; Height, 0–4 nm. Middle, Histograms of the distances between two globular lobes from 194 frames of 1 molecule. Representative images of closed and open conformations with height profiles are shown right. Bottom, Height profiles of simulated AFM images from the crystal structure (PDB ID: 1UMK) (left) and the manually arranged structure (right) with the simulated images. **c** The closed conformation (left) corresponds to the crystal structure (1UMK) while the open conformation (right) was manually prepared based on the HS-AFM data. CYB5R3 and FAD are shown with surface and space-filling models, respectively. **d** Color comparison of purified free CYB5R3 with ufmylated CYB5R3. Recombinant CYB5R3 and UFM1-conjugated CYB5R3 were purified in large quantities as described in Fig. S6. **e** In vitro enzymatic activity of recombinant CYB5R3 ($n = 3$) and UFM1-conjugated CYB5R3 ($n = 3$) were measured as described in Methods section. Data are means ± s.e. Statistical analysis was performed by Šidák's multiple comparison test after one-way ANOVA. **f** Left panel: $CYB5R3^{-/-}$ $UFSP2^{-/-}$ HEK293T expressing indicated constructs were lysed and then subjected to immunoblot analysis with anti-CYB5R3 antibody. Data shown are representative of three separate experiments. Right panel: the reductase activity of microsomal fractions prepared from the aforementioned cells was measured as described in the Methods section. Data are means ± s.e. Statistical analysis was performed by Šidák's multiple comparison test after one-way ANOVA. **g, h** The amounts of saturated, monounsaturated, and polyunsaturated phosphatidylcholine and of phosphatidylethanolamine (**g**) ($n = 3$) and of cholesterol (**h**) ($n = 4$) in $CYB5R3^{-/-}$ $UFSP2^{-/-}$ HEK293T expressing UFM1, UFL1, UFBP1, and CYB5R3 or CYB5R3$^{K214R}$. Data are means ± s.e. Source data are provided as a Source Data file.

conjugation site for UFM1, was positioned at the interface between the NADH and FAD domains, its ufmylation would bias the equilibrium between closed and open conformations toward the open one, leading to the destruction of the FAD-binding groove and thereby reducing the enzymatic activity of CYB5R3.

When large amounts of recombinant UFM1, UBA5, and UFC1 were mixed with a lysate of *E. coli* expressing CYB5R3ΔN26, but not recombinant CYB5R3ΔN26, CYB5R3ΔN26 was ufmylated (Supplementary Fig. 6). Although this appears to be an artificial reaction system that does not require E3, we utilized the ufmylated CYB5R3ΔN26 purified from this system. Remarkably, while free purified CYB5R3 exhibited a brilliant yellow color, the conjugated form was clear (Fig. 3d). Since FAD emits yellow-green fluorescence, these results indicate that FAD is released from CYB5RC by UFM1 conjugation. FAD is a co-factor that plays a vital role in many FAD-dependent enzymatic reactions, including CYB5R3[27,28]. Thus, the conjugated form completely lost its reductase activity (Fig. 3e). The reductase activity of the microsomal fraction of *CYB5R3*−/− *UFSP2*−/− HEK293T cells was significantly lower than that of parental HEK293T cells, and it was recovered by sole expression of wild-type or mutant CYB5R3 (Fig. 3f). When the E3 components were co-expressed with wild-type CYB5R3, the reductase activity decreased (Fig. 3f). This decrease was not observed in the CYB5R3[K214R] mutant (Fig. 3f), suggesting that ufmylation negatively regulated the enzymatic activity of CYB5R3 even in vivo. Considering that CYB5R3 has two forms (Fig. 3b), ufmylation may direct CYB5R3 toward its inactive, open form. Since CYB5R3 is a reductase involved in fatty acid elongation and desaturation and cholesterol biogenesis[25], we expected increased levels of unsaturated fatty acids in *CYB5R3* knockout cells harboring the ufmylation-defective CYB5R3[K214R] mutant. However, comprehensive lipidome analyses showed that the level of unsaturated fatty acids in *CYB5R3* and *UFSP2* double-knockout cells expressing wild-type CYB5R3 was comparable to that in cells expressing CYB5R3[K214R] (Fig. 3g and Supplementary Data 3). We also confirmed a comparable level of cholesterol between *CYB5R3* and *UFSP2* double-knockout cells expressing wild-type CYB5R3 and CYB5R3[K214R] (Fig. 3h). Taken together, these data indicate that ufmylation of CYB5R3 decreases its enzymatic activity, but the defect barely affects the synthesis of unsaturated fatty acids or cholesterol, probably due to compensatory function by other proteins in this family, namely CYB5R1, 2, and 4.

### The association of ufmylated CYB5R3 with UFBP1

An immunoprecipitation assay revealed that wild-type CYB5R3, but not CYB5R3[K214R], interacted with UFBP1 (Fig. 4a). Similarly, immunoprecipitants of wild-type CYB5R3, but not of CYB5R3[K214R], contained UFL1 (Fig. 4a). An in vitro pull-down assay showed that while ufmylated CYB5R3ΔN26 was pulled down with GST-UFBP1, free CYB5R3 was not (Fig. 4b). This assay also demonstrated a direct interaction between free UFM1 and UFBP1 (Fig. 4b). These results imply that UFBP1 has the ability to bind to not only free UFM1 but also to UFM1-conjugated CYB5R3, at least in vitro. To determine which region of UFBP1 is required for the interaction with UFM1, we prepared a deletion series of UFBP1 (M1–M5) and performed a pull-down assay (Fig. 4c). This revealed that UFBP1 M4, which covers amino acids 116–214 and is conserved across species (Fig. 4d), is essential and sufficient for the interaction between UFBP1 and UFM1 (Fig. 4c). As reported elsewhere[12], UFL1 interacted with UFBP1 M5 containing the PCI domain (Fig. 4c). Analysis of the secondary structure of UFBP1 M4 revealed that the region comprises a coiled-coil motif, a β-strand, and an α-helix, and the β-strand contains FVVEE (Fig. 4d); overall this structure resembles the UFM1-interacting motifs (UFIMs) of UBA5[29] and STK38[30]. We crystallized UFM1 in a complex with the UFIM of UFBP1 (aa 194–202) by fusing the UFIM at the N-terminus of UFM1, and determined the crystal structure at a resolution of 1.6 Å (Supplementary Fig. 7a and Table 1). In the crystal, UFBP1 UFIM bound to an adjacent UFM1 in a head-to-tail

manner (Supplementary Fig. 7b); a single UFM1–UFBP1 UFIM complex is shown in Fig. 4e, left. UFBP1 UFIM has an extended β-conformation and forms an intermolecular, anti-parallel β-sheet with the β2 of UFM1, which is similar to the way that UBA5 UFIM interacts with UFM1[29] (Fig. 4e, right). Further, three hydrophobic residues of UBA5 UFIM, namely Trp341, Ile343, and Leu345, engage in hydrophobic interactions with UFM1. Phe196 and Val198 of UFBP1 UFIM are positioned equivalently to Ile343 and Leu345 of UBA5, respectively (Fig. 4f), and demonstrate hydrophobic interactions with UFM1, whereas no residue corresponds to UBA5 Trp341 (Fig. 4e). In addition to hydrophobic interactions, UFBP1 Glu199 forms hydrogen bonds with the side chains of Lys3 and Ser22, which likely play an auxiliary role in increasing binding affinity. An in vitro pull-down assay showed that a UFBP1 mutant with point mutations at Phe196 and Val198 (UFBP1[F196A V198A]) hardly bound to either free UFM1 or ufmylated CYB5R3 (Fig. 4g). As expected, an immunoprecipitation assay showed that UFBP1[F196A V198A] had considerably reduced ability to bind to ufmylated CYB5R3 (Fig. 4h).

We next investigated whether the UFBP1 UFIM mutant exhibited E3 activity in cells. To exclude the effect of endogenous UFBP1, we generated *UFBP1* knockout HEK293T cells (Supplementary Fig. 2). An immunoprecipitation assay showed that both MYC-tagged wild-type UFBP1 and UFBP1[F196A V198A] bound to endogenous UFL1 (Fig. 4i). Co-expression of wild-type UFBP1 or UFBP1[F196A V198A] with UFL1 increased the ufmylation of endogenous CYB5R3 (Fig. 4j). Remarkably, the level of ufmylated CYB5R3 in cells harboring UFBP1[F196A V198A] was lower than that in cells expressing wild-type UFBP1 (Fig. 4j). Since the ligase activity of the UFL1–UFBP1 complex is known to be increased by UFBP1 ufmylation at lysine 267[31], we next investigated the ufmylation level of UFBP1. While co-expression of UFL1 with wild-type UFBP1 induced UFBP1 ufmylation, as reported previously[6], hardly any ufmylation occurred when UFL1 was co-expressed with UFBP1[F196A V198A] (Fig. 4j). Unlike UFBP1, the ufmylation-defective UFBP1[K267R] mutant did not demonstrate enhanced E3 activity against CYB5R3 (Fig. 4k). These results suggest that the association of ufmylated CYB5R3 with UFBP1 enhances UFL1–UFBP1 ligase activity through UFBP1 ufmylation.

### Ufmylation of CYB5R3 becomes a signal for an ER-phagy

We next sought to determine the physiological significance of ufmylated CYB5R3. We first explored the possibility that it is involved in ER-phagy, since genome-wide CRISPR screening identified UFL1 and UFBP1 as activators of ER-phagy[23]. First, we utilized an ER-phagy reporter containing an N-terminal ER signal sequence followed by tandem monomeric RFP and GFP sequences and the ER retention sequence KDEL (ssRFP-GFP-KDEL). With this reporter, ER-phagy activity can be monitored by fluorescence microscopy[32]. The reporter should appear yellow (green and red) in the ER matrix. When it is transported to lysosomes by autophagy, it becomes red, because GFP, but not RFP, is quickly quenched in the acidic environment. Hence, the total RFP intensity of these red punctae should indicate the amount of the ER-phagy reporter delivered to lysosomes. In *CYB5R3*-knockout HeLa cells expressing ssRFP-GFP-KDEL together with UFL1, UFBP1 and wild-type CYB5R3, the number of RFP-positive punctae increased under nutrient-deprived conditions (Supplementary Fig. 8). Such induction was observed even in the case of expression of CYB5R3[K214R] (Supplementary Fig. 8). There is a possibility that since ufmylated CYB5R3 is involved in ER-phagy for restricted ER subpopulation, ssRFP-GFP-KDEL that locates in whole ER is unable to monitor the ufmylated CYB5R3-mediated ER-phagy. To this end, we monitored the lysosomal degradation of the CYB5R3 reporter fused with tandem mCherry and GFP sequences (CYB5R3-CG) (Fig. 5a). When UFL1, UFBP1, and CYB5R3-CG were expressed in *CYB5R3* knockout HeLa cells, the number of mCherry-positive punctae increased under nutrient-deprived but not nutrient-rich conditions (Fig. 5b). Most of the ER-phagy pathways that have been identified thus far are induced by

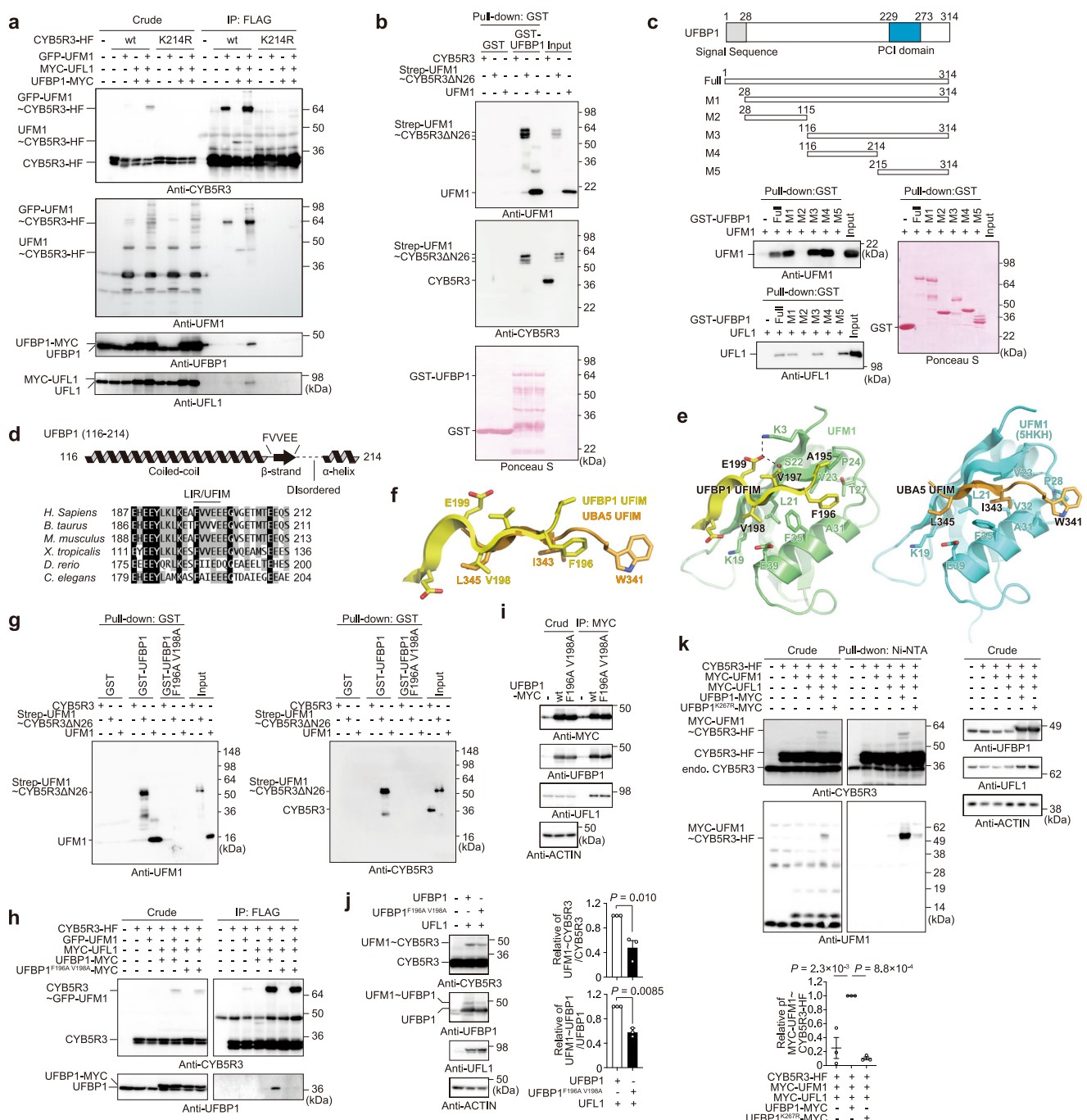

**Fig. 4 | Ufmylated CYB5R3 interacts with UFBP1. a** *CYB5R3⁻/⁻ UFSP2⁻/⁻* HEK293T expressing indicated constructs were lysed and then subjected to immunoprecipitation. **b–c** In vitro pull-down assay. The amount of each protein bound to GST-UFBP1 was estimated by immunoblot (**b**). The faster-migrating ufmylated CYB5R3ΔN26 proteins are most likely degradation products due to long-term storage and freeze–thawing. UFM1 and UFL1 binding to GST-UFBP1 and each mutant was estimated by immunoblot (**c**). **d** Top, analysis of the secondary structure of UFBP1 (116–214) using PSIPRED/DISOPRED[61,62] and COILS[63]. Bottom, alignment of UFIM of UFBP1. **e** Crystal structure of UFM1 complexed with the UFIM of UFBP1 (left) or the UFIM of UBA5 (right, PDB 5HKH). The side chains of the residues involved in the UFM1-UFIM interaction are shown with a stick model. Broken lines indicate possible hydrogen bonds. **f** Structural comparison between UFBP1 and UBA5 UFIMs complexed with UFM1. The figure was prepared by superimposing the structure of the UFIM moiety of UFBP1-UFM1 onto that of UBA5 UFIM-UFM1. **g** In vitro pull-down assay. The amount of each protein bound to GST-UFBP1 was

estimated by immunoblot. **h–i** Wild-type (**h**) and *UFBP1⁻/⁻* HEK293T (**i**) expressing indicated constructs were lysed and then subjected to immunoprecipitation. **j** HEK293T expressing indicated constructs were lysed and then subjected to immunoblot. Bar graphs show the relative value of the ratio of endogenous ufmylated CYB5R3 to CYB5R3 was 1 and the relative value of the ratio of ufmylated UFBP1 to UFBP1 as 1 when UFL1 and UFBP1 were overexpressed, respectively. Data are means ± s.e. Statistical analysis was performed by two-sided Welch's *t* test. **k** HEK293T expressing indicated constructs were lysed and then subjected to pull-down assay with Ni-NTA agarose under denaturing conditions, followed by immunoblot. Bar graph shows the relative value of the ratio of MYC-UFM1-conjugated CYB5R3-His-FLAG to CYB5R3-His-FLAG in crude lysates as 1 when UFL1 and UFBP1 were overexpressed. Data are means ± s.e. Statistical analysis was performed by Šidák's multiple comparison test after one-way ANOVA. Data for immunoblots presented in this Figure are representative of three separate experiments. Source data are provided as a Source Data file.

## Table 1 | Data collection and refinement statistics

| | Human CYB5R3 | UFBP1 UFIM-UFM1 |
|---|---|---|
| Data collection | | |
| Space group | *C*2 | *P*3$_1$21 |
| Cell dimensions | | |
| *a*, *b*, *c* (Å) | 163.36, 79.92, 169.40 | 53.41, 53.41, 82.23 |
| α, β, γ (°) | 90, 98.72, 90 | 90, 90, 90 |
| Resolution (Å) | 49.56–2.46 (2.61–2.46) | 27.41–1.60 (1.63–1.60) |
| $R_{merge}$ | 0.259 (1.78) | 0.036 (0.81) |
| I/σI | 7.21 (0.89) | 34.9 (3.0) |
| Completeness (%) | 100.0 (100.0) | 100.0 (100.0) |
| Redundancy | 12.5 (11.8) | 11.8 (7.1) |
| Refinement | | |
| Resolution (Å) | 49.56–2.46 | 26.71–1.60 |
| No. of reflections | 78626 | 18455 |
| $R_{work}$/$R_{free}$ | 0.2312/0.2595 | 0.1808/0.2000 |
| No. of atoms | | |
| Protein | 10761 | 680 |
| Ligand/ion | 288 | |
| Water | 192 | 151 |
| *B*-factors | | |
| Protein | 69.7 | 25.6 |
| Ligand/ion | 65.1 | |
| Water | 56.9 | 41.4 |
| R.m.s. deviations | | |
| Bond lengths (Å) | 0.014 | 0.016 |
| Bond angles (°) | 1.06 | 1.73 |

Number of xtals used is six (CYB5R3) and one (UFBP1 UFIM-UFM1).
Values in parentheses are for the highest-resolution shell.

nutrient starvation and ER stress[32–34], implying that ER-phagy induction requires a priming effect that activates autophagy-related proteins. Similarly, ER-phagy by UFM1 probably also requires such a priming effect. The increase in punctae upon nutrient deprivation was completely abolished by treatment with Bafilomycin A$_1$, an inhibitor of lysosomal acidification (Fig. 5b). The lysosomal delivery of CYB5R3-CG was also markedly suppressed by the substitution of Lys214 with Arg (Fig. 5b). Likewise, in *UFBP1*-deficient HeLa cells harboring UFL1 and UFBP1$^{F196A\ V198A}$ or UFBP1$^{K267R}$, the amount of CYB5R3-CG delivered to lysosomes was much smaller than that in cells expressing wild-type UFL1 and UFBP1 (Fig. 5c). There are two major ways to deliver ER to lysosomes: macro-ER-phagy, which is associated with autophagosome formation around the targeted ER subdomain, and micro-ER-phagy, wherein lysosomes invaginate and surround the targeted ER subdomain[35,36]. Immunofluorescence microscopy revealed that punctae positive for CYB5R3-GFP were hardly observed under nutrient-rich conditions, but upon nutrient deprivation, they increased in number and co-localized with core ATG machineries, such as FIP200, WIPI2, and LC3, that are essential for autophagosome formation (Fig. 5d). This suggests the involvement of macro-ER-phagy in the degradation of CYB5R3. In fact, the loss of *ATG7*, a core *ATG* gene, suppressed CYB5R3-CG translocation from the ER membrane to lysosomes under starvation conditions even when UFL1 and UFBP1 were overexpressed (Fig. 5e). Taken together, we concluded that ufmylated CYB5R3 is degraded by macro-ER-phagy.

### CDK5RAP3 is required for the ufmylation-mediated ER-phagy

It was recently proposed that CDK5RAP3, which binds to both UFL1 and autophagosome-localizing ATG8 family proteins, functions as both a substrate adaptor that directs ufmylation to the ribosomal protein RPL26[8,9], and as an adaptor protein for ufmylation-dependent

ER-phagy[34]. We confirmed the interaction of endogenous CDK5RAP3 and UFL1 by an immunoprecipitation assay (Supplementary Fig. 9a). Anti-CDK5RAP3 antibody revealed neither endogenous GABARAP- nor LC3-family proteins in the immunoprecipitants (data not shown), probably due to the low sensitivity of our immunoprecipitation assay. We, therefore, used a Fluoppi system, which is a fluorescence-based technology to detect protein-protein interactions in living cells with a high signal-to-noise ratio[37,38]. An Ash tag, which forms a homo-oligomer, was fused to LC3 and GABARAP, and a homotetrameric humanized Azami-Green (hAG) tag was fused to CDK5RAP3. We co-expressed Ash-LC3 or Ash-GABARAP with hAG-CDK5RAP3 in HEK293T cells and verified that their expression was comparable to endogenous levels (Supplementary Fig. 9b). If multivalent interactions between Ash-LC3 or Ash-GABARAP and hAG-CDK5RAP3 occur, hAG forms phase-separated fluorescent foci in cells. We observed Fluoppi foci consisting of hAG-tagged CDK5RAP3 and Ash-tagged LC3 and GABARAP (Supplementary Fig. 9c). The Fluoppi foci were not observed with the expression of the hAG-tagged CDK5RAP3 LC3-interacting region-mutant, in which Trp269, Trp294, and Trp312 were substituted with Ala (CDK5RAP3$^{3WA}$)[34] (Supplementary Fig. 9c). An in vitro pull-down assay showed that recombinant CDK5RAP3 interacted not only with UFL1, but also with GABARAP- and LC3-family proteins, with the exception of LC3C (Supplementary Fig. 9d, e).

When GFP-UFM1 alone or in various combinations with UFL1, UFBP1, and CDK5RAP3 was expressed in *UFSP2* knockout HEK293T cells, the level of endogenous CYB5R3 conjugated with GFP-UFM1 was increased by UFL1 expression and by the co-expression of UFL1 and UFBP1 (Fig. 6a). This up-regulation was canceled by concomitant expression of CDK5RAP3 (Fig. 6a). UFBP1 conjugated with one or a few UFM1 molecules showed a similar pattern as ufmylated CYB5R3 (Fig. 6a). To clarify whether the decreased levels of ufmylated CYB5R3 and UFBP1 caused by CDK5RAP3 expression were due to macro-ER-phagy, we conducted experiments with Bafilomycin A$_1$. To elucidate the role of CDK5RAP3 in the lysosomal degradation of ufmylated CYB5R3, we generated *CDK5RAP3* and *UFSP2* double-deficient HEK293T cells (Supplementary Fig. 2). The expression of the E3 components in *CDK5RAP3*$^{-/-}$ *UFSP2*$^{-/-}$ HEK293T cells increased the amounts of GFP-UFM1-conjugated CYB5R3 and UFBP1 (Fig. 6b), and the amounts of both were decreased by simultaneous expression of CDK5RAP3 (Fig. 6b). In the absence of CDK5RAP3, Bafilomycin A$_1$ treatment did not affect the amounts of either GFP-UFM1-conjugated CYB5R3 or UFBP1 (Fig. 6b). In contrast, in the presence of CDK5RAP3, the amounts of both were partially recovered by Bafilomycin A$_1$ treatment (Fig. 6b). The fact that this recovery was partial and not full may be due to inhibitory ligase activity of CDK5RAP3[9]. In agreement with these results, an assay using the CYB5R3-CG reporter revealed that the transport of CYB5R3 from the ER membrane to lysosomes, which was induced by starvation and the overexpression of UFL1 and UFBP1, was abolished by ablation of CDK5RAP3 (Fig. 6c). This suppression was canceled by the expression of CDK5RAP3 (Fig. 6c). These results indicate that ufmylated CYB5R3 is degraded in macro-ER-phagy in a CDK5RAP3-dependent fashion.

### Ufmylation-defective *Cyb5r3* knock-in mice show microcephaly

Finally, to examine the physiological importance of CYB5R3 ufmylation in vivo, we developed *Cyb5r3*$^{K214R/K214R}$ knock-in mice in which Cyb5r3 ufmylation was disrupted (Supplementary Fig. 10). *CYB5R3* is known to cause recessive congenital methemoglobinemia (RCM) types I and II[39]. Type I RCM is characterized by a deficiency of the soluble isoform and manifests as cyanosis of the skin and mucous membranes. In type II, the defect affects both soluble and membrane-bound isoforms and therefore influences all body tissues, including red blood cells and leukocytes. This type is associated with severe encephalopathy with mental retardation, microcephaly, generalized

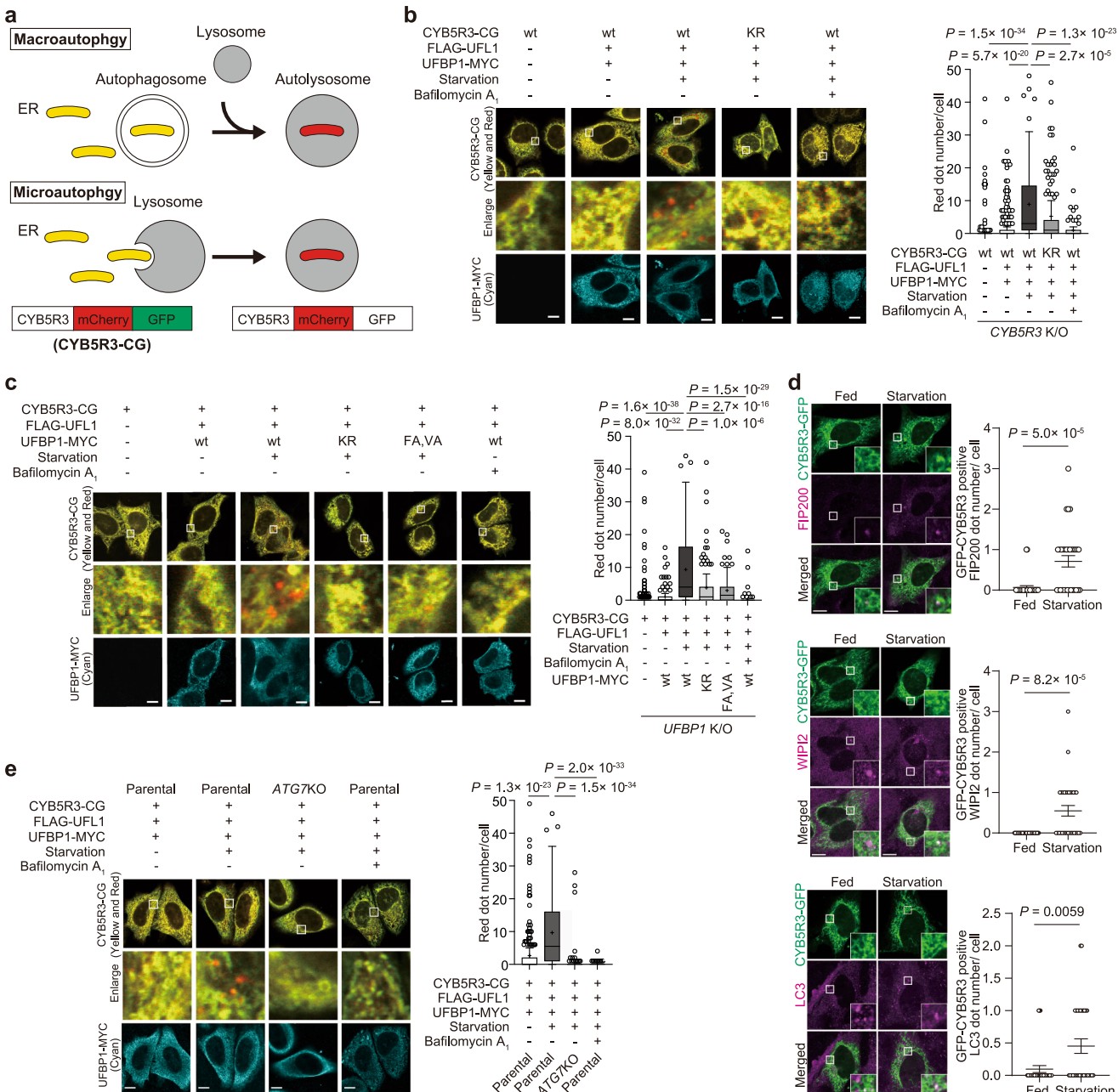

**Fig. 5 | The regulation of ER-phagy by ufmylation of CYB5R3. a** Schematic representation of the CYB5R3-mCherry-GFP (CYB5R3-CG) reporter for monitoring lysosomal degradation of CYB5R3. **b** Fluorescence microscopic analysis. CYB5R3-CG or CYB5R3^{K214R}-CG (KR) together with UFL1 and UFBP1 were co-transfected into *CYB5R3*^{−/−} HeLa. Forty-eight hours after transfection, the cells were cultured under nutrient-rich or deprived conditions for 9 h in the presence or absence of Bafilomycin A_1. The cells were fixed and observed by confocal microscopy. The number of single mCherry-positive punctae per cell was determined using a Benchtop High-Content Analysis System and CellPathfinder software without bias. The numbers of cells used to count the mCherry-positive punctae were 401, 226, 117, 136, and 129 from lanes 1 to 5. Bars: 10 μm. **c** Fluorescence microscopic analysis. CYB5R3-CG together with UFL1 and UFBP1 or UFBP1^{F196A VI98A} (FA, VA) were co-transfected into *UFBP1*^{−/−} HeLa. The cells were analyzed as described in **b**. The numbers of cells used to count the mCherry-positive punctae were 335, 176, 90, 123, 114, and 102 from lanes 1 to 6. Bars: 10 μm. **d** Fluorescence microscopic analysis. CYB5R3-GFP together with UFL1 and UFBP1 were co-transfected into *CYB5R3*^{−/−} HeLa. Forty-eight hours after transfection, the cells were cultured under nutrient-rich or deprived conditions for 9 h. The cells were fixed, immunostained with anti-FIP200, anti-WIPI2, and anti-LC3 antibodies, and observed by confocal microscopy. Bars: 10 μm. The number of punctae positive for GFP and FIP200, WIPI, or LC3 per cell (*n* = 30) was determined. **e** Fluorescence microscopic analysis. CYB5R3-CG together with UFL1 and UFBP1 were co-transfected into parental or *ATG7*-deficient HeLa cells. The cells were observed as described in **b**. The numbers of cells used to count the mCherry-positive punctae were 245, 150, 186, and 144 from lanes 1 to 4. Bars: 10 μm. For the box plots, horizontal bars indicate medians, boxes the interquartile range (25th–75th percentiles), and whiskers 1.5× the interquartile range; outliers are plotted individually. Data are means ± s.e. Statistical analysis was performed by Šidák's multiple comparison test after one-way ANOVA. Source data are provided as a Source Data file.

dystonia, and movement disorders[39,40]. This clinical picture is similar to that of patients with mutations of *UBA5, UFC1,* or *UFM1*[16–18]. Thus, we focused on neurological studies in the mutant mice. Macroscopic anatomical analysis of the brains of 4.5-month-old *Cyb5r3*^{K214R/K214R} mice

revealed microcephaly (Fig. 6d). The transverse length of the mutant brains, but not the longitudinal length, was significantly shorter than that of wild-type and *Cyb5r3*^{K214R/+} brains (Fig. 6d). Histological analysis using hematoxylin and eosin staining showed no apparent abnormality

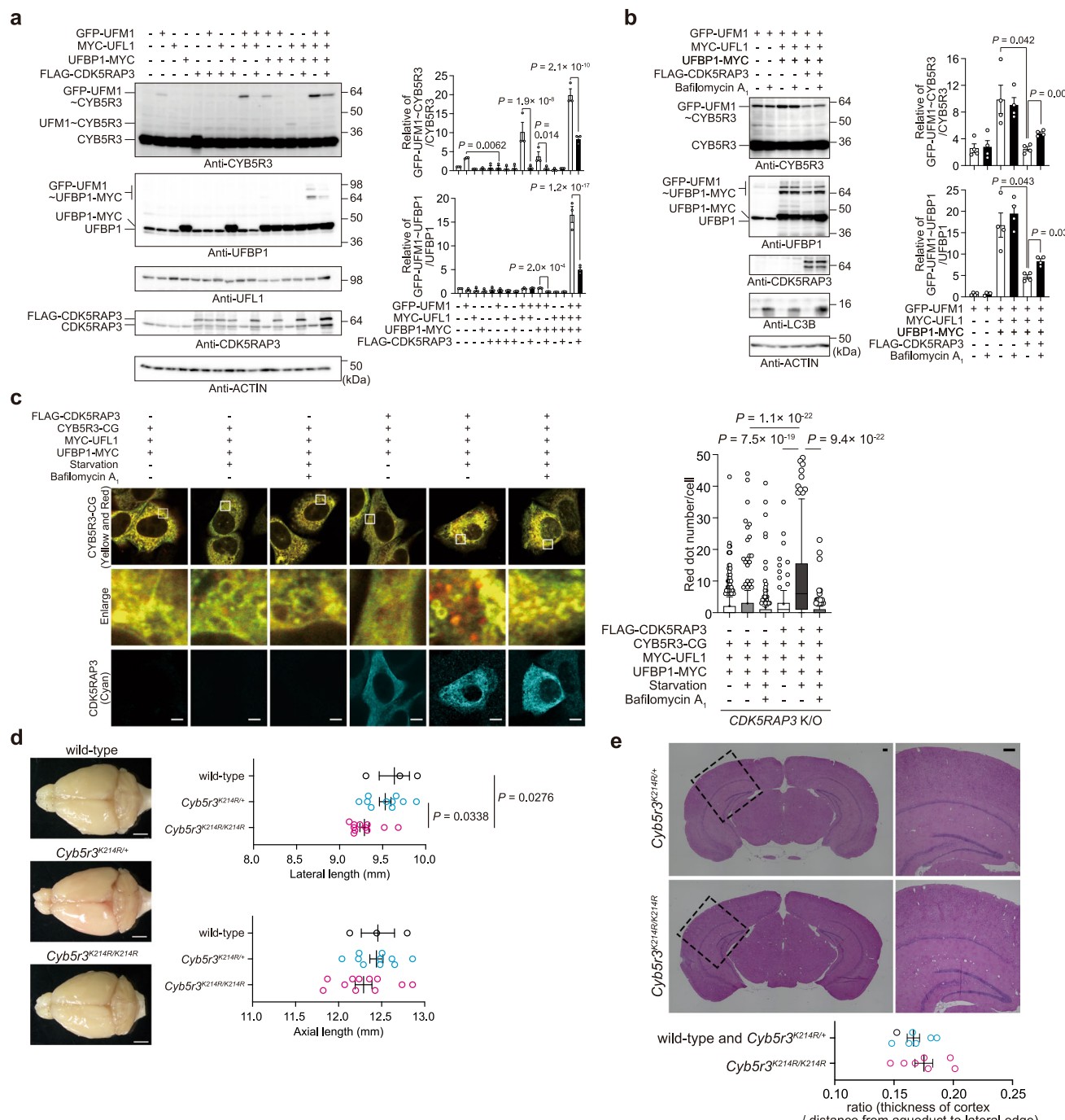

**Fig. 6 | CDK5RAP3 is indispensable for ufmylation-mediated ER-phagy.**
**a**−**b** *UFSP2*−/− (**a**) or *CDK5RAP3*−/− *UFSP2*−/− HEK293T (**b**) expressing indicated constructs were cultured under nutrient-rich conditions (**a**) in the presence or absence of Bafilomycin A₁ for 12 h (**b**). Lysates were subjected to immunoblot analysis. Data shown are representative of three separate experiments. Bar graphs show the results of quantitative densitometric analysis of the indicated proteins relative to free CYB5R3 or free UFBP1. Data are means ± s.em. Statistical analysis was performed by Šidák's multiple comparison test after one-way ANOVA. **c** Fluorescence microscopic analysis. *CDK5RAP3*−/− HeLa expressing indicated constructs were analyzed as described in Fig. 5b. The numbers of cells used to count the mCherry-positive punctae were 337, 235, 191, 127, 113, and 92 from lanes 1 to 6. Bars: 10 μm. Horizontal bars indicate medians, boxes in the interquartile range (25th–75th percentiles), and whiskers 1.5× the interquartile range; outliers are plotted individually. Data are means ± s.e. Statistical analysis was performed by Šidák's multiple

comparison test after one-way ANOVA. **d** Dorsal view of the brains of 4-month-old wild-type, *Cyb5r3^K214R/+*, and *Cyb5r3^K214R/K214R* mice. Graphs show the axial length and the maximal lateral length of brains of mice of the indicated genotypes. Data are means ± s.e. of wild-type (*n* = 3), *Cyb5r3^K214R/+* (*n* = 10) and *Cyb5r3^K214R/K214R* (*n* = 11) mice. Statistical analysis was performed by Šidák's multiple comparison test after one-way ANOVA. Bars: 2 mm. **e** H&E staining of brain sections from 4.5–5 month-old *Cyb5r3^K214R/+* and *Cyb5r3^K214R/K214R* mice. Boxed regions are magnified and shown on the right. The graph shows ratios of the thickness of the auditory cortex to the distance from the aqueduct to the corresponding lateral edge. Data are means ± s.e. of wild-type (*n* = 1 from one mouse, black) and *Cyb5r3^K214R/+* (*n* = 6 from three mice, blue), and *Cyb5r3^K214R/K214R* (*n* = 7 from four mice, pink) mice. Statistical analysis was performed on the wild-type and *Cyb5r3^K214R/+* mice together as controls, and these were compared to the *Cyb5r3^K214R/K214R* mice. Bars: 2 mm. Source data are provided as a Source Data file.

in cellular organization in the mutant brain (Fig. 6e). These results indicate that defective ER-phagy mediated by CYB5R3 ufmylation causes atrophy of the whole brain rather than of specific regions.

## Discussion

Here we demonstrated that (1) an ER membrane-localizing reductase, CYB5R3, is a bona fide UFM1 substrate; (2) ufmylated CYB5R3 interacts with the E3 ligase component UFBP1 to facilitate further ufmylation of CYB5R3; and (3) CYB5R3 ufmylation induces CDK5RAP3-mediated macro-ER-phagy.

In macro-ER-phagy, isolation membranes/phagophores form along ER marked for degradation. ER-localizing adaptor proteins, which demonstrate affinity for ATG8 family proteins and/or FIP200, ensure autophagy selectivity against the ER[35,36]. To date, seven ER-phagy adaptors have been identified in mammals: FAM134B[33], RTN3L[41], ATL3[42], SEC62[43], CCPG1[44], TEX264[32,45], and CALCOCO1[46]. While these adaptors may play redundant roles in ER-phagy, they are expressed in specific tissues, localize on distinct ER subdomains such as sheet ER, tubular ER and three-way junctions and degrade separate subdomains of the ER and different ER proteins[35,36]. Therefore, it is plausible that each ER-phagy adaptor participates in spatiotemporal regulation and/or tissue-specific cellular functions. Both UFL1 and UFBP1 play an important role in autophagic degradation of sheet ER[23]. Supporting this, UFBP1 expression has been shown to trigger ER expansion, occasionally forming organized smooth ER substructures[11]. In addition to ufmylation of the ribosomal protein RPL26, which is a signal for ER-phagy[23,34], we propose that CYB5R3 ufmylation, which is induced by UFL1 and UFBP1, serves as a signal for macro-ER-phagy (Supplementary Fig. 11). First, CYB5R3 is ufmylated, and is thereafter recognized by UFBP1. Second, this interaction may promote UFBP1 ufmylation and increase the E3-ligating activity of UFL1–UFBP1 against CYB5R3. Finally, UFL1 in complex with ufmylated CYB5R3 and UFBP1 interacts with CDK5RAP3, leading to autophagic degradation of ER subdomains. In addition, while CDK5RAP3 facilitates the ufmylation of RPL26, it may suppress the ufmylation of other substrates, including CYB5R3 and UFBP1[8,9]. In other words, under conditions that do not result in ufmylation of RPL26, ufmylation of other substrates such as CYB5R3 may be promoted. While RPL26 ufmylation is involved in ER-phagy under ribosomal stalling conditions[22], that of CYB5R3 may participate in ER-phagy in different conditions. Since both UFL1 and UFBP1 are ER stress-inducible proteins[47], the ER-phagy mediated by CYB5R3 ufmylation may play an important role in stress conditions.

CYB5R3 mutations have clinical manifestations that resemble those of mutations of genes encoding ufmylation-related components. Biallelic mutations of UBA5 cause severe, early-onset developmental disorders[16,17] that manifest during early infancy as severe irritability, followed by dystonia and impaired development. Furthermore, the majority of individuals display postnatal microcephaly and epilepsy and develop spasticity. Biallelic mutations of UFM1 or UFC1 are also associated with microcephaly, global developmental delay, and seizures[18,48]. RCM type II caused by CYB5R3 mutations is characterized by severe neurological symptoms similar to those of patients with mutations of UBA5, UFC1, or UFM1[40]. Since the membrane-bound form of CYB5R3 exists mainly on the cytoplasmic side of the ER and plays roles in desaturation and elongation of fatty acids, cholesterol biosynthesis, and drug metabolism, abnormal lipid metabolism related to fatty acid desaturation has been proposed as a primary cause of RCM type II. However, a growing body of evidence on CYB5R3 functions has shown that defective lipid metabolism is probably a secondary phenotype rather than a direct cause of the disease[40]. In this study, we generated ufmylation-defective Cyb5r3[K214R/K214R] knock-in mice and found that these mice exhibited microcephaly (Fig. 6d, e). Though we do not exclude a possibility that the mutation inhibits other post-translational modification(s) except for the ufmylation, our results suggest that the defect in macro-ER-phagy through the ufmylation of CYB5R3 is involved in the pathogenesis of RCM type II.

## Methods

### Cell culture

HEK293T (ATCC CRL-3216) and HeLa (ATCC CCL2) cells were grown in Dulbecco's modified Eagle medium (DMEM) containing 10% fetal bovine serum (FBS), 2 mM L-glutamine, 5 U/ml penicillin, and 50 µg/ml streptomycin. For overexpression experiments, HEK293 and HeLa cells were transfected with PEI MAX (Polysciences, Inc., Warrington, PA, USA) and Lipofectamine 3000 (Thermo Fisher Scientific, Waltham, MA, USA), respectively. For knockdown experiments, HEK293T cells were transfected with 25 nM SMARTpool siRNA for UFL1 (M-014027-01-0005, Horizon Discovery, Cambridge, UK) or for UFBP1 (M-014256-00-0005, Horizon Discovery) using Dharmafect 1 (Horizon Discovery). To generate CYB5R3 UFSP2 double-knockout, UFBP1 knockout, CYB5R3-knockout, and CDK5RAP3-knockout cells, CYB5R3 (5′-AGC CATCACCCTCGAGAGCC-3′), UFBP1 (5′-GTAGCGGCGGCTCTGCTAGT-3′) or CDK5RAP3 (5′-TGCTGCGGTTCGTGCAGAAG-3′) guide RNA designed using the CRISPR Design tool (http://crispr.mit.edu/) were subcloned into pX330-U6-Chimeric_BB-CBh-hSpCas9 (Addgene #42230), a human codon-optimized SpCas9 and chimeric guide RNA expression plasmid. HEK293T, UFSP2 knockout HEK293T[12] or HeLa cells were co-transfected with vectors pX330 and pEGFP-C1 (#6084-1, Clontech Laboratories, Mountain View, CA, USA) and cultured for 2 d. Thereafter, GFP-positive cells were sorted and expanded. Loss of CYB5R3 or UFBP1 was confirmed by a heteroduplex mobility assay followed by immunoblot analysis with anti-CYB5R3 or anti-UFBP1 antibody. UFSP2-deficient HEK293T[12] and ATG7-deficient HeLa[49] cells were used in this study. HEK293T and HeLa cells were authenticated by STR profile. All cell lines were tested for mycoplasma contamination.

### Generation of Cyb5r3[K214R/K214R] knock-in mice

To generate knock-in mice by CRISPR/Cas technology, CRISPR RNA (crRNA) was designed to recognize exon8 of Cyb5r3 (primer: 5′-AAG-GATGGCCGTATCGTCAG-3′). Synthetic crRNA (Alt-R™ CRISPR-Cas9 crRNA XT), tracrRNA (Alt-R™ CRISPR-Cas9 tracrRNA), and Cas9 protein (Alt-R™ S.p. Cas9 Nuclease V3) were purchased from Integrated DNA Technologies, Inc. (IDT) (Coralville, IA, USA). A single-stranded oligodeoxynucleotide (ssODN) (5′-CCAAGGGCAGTCCCTGAGTCACCT GTCTTGACTGGGTGGCCTGTCTCCTGCAGTCCGAGAGAGACATCCTG TTGCGGCCTGAGCTGGAGGAACTGAGGAACGAACATTCTGCTCGCTT CAAGCTC-3′) harboring a K214R mutation flanked by 60-bp homologous arms corresponding to exon8 of Cyb5r3 was custom synthesized by Eurofins Genomics Japan (Tokyo, Japan) for CRISPR/Cas-mediated knock-in. The CRISPR/Cas9 solution was prepared as previously described[50], with minor modifications. Briefly, lyophilized crRNAs and tracrRNA were resuspended in Nuclease-Free Duplex Buffer (IDT) to a concentration of 240 µM. Equal volumes of crRNA and tracrRNA were combined, heated at 95 °C for 5 min, and then placed at room temperature (RT) for -10 min to allow for the formation of crRNA:tracrRNA duplex, which was mixed with Cas9 protein to form a ribonucleoprotein (RNP) complex. Lyophilized ssODN was resuspended in nuclease-free water to a concentration of 4 µg/µl. The RNP complex was mixed with ssODN and diluted in Opti-MEM (Thermo Fisher Scientific, Waltham, MA, USA). The final concentrations of Cas9 protein, crRNA:tracrRNA duplex, and ssODN were 1 µg/µl, 30 µM and 1 µg/µl, respectively. To induce CRISPR/Cas-mediated knock-in, -1.5 µl of CRISPR/Cas9 solution was injected into the oviductal lumens of female C57BL/6 N mice at day 0.7 of pregnancy. Immediately after the injection, the oviduct regions were grasped with a tweezer-type electrode (CUY652-3; Nepa Gene, Chiba, Japan) and then electroporated using a NEPA21 square-wave pulse generator (Nepa Gene). The electroporation parameters used were as follows: poring pulse: 45 V, 5 ms, 50-ms interval, 3 pulses, 10% decay (± pulse orientation); and transfer pulse: 10 V, 50 ms, 50-ms

interval, 3 pulses, 40% decay (± pulse orientation). Pregnant female mice were allowed to deliver their pups. All mice housed in a specific pathogen-free room under temperature (23 ± 3 degrees), and humidity (40-60%) controlled conditions with 12/12 h light-dark cycle. The Ethics Review Committee for Animal Experimentation of Juntendo University approved the experimental protocol, and that we have complied with all relevant ethical regulations.

## Immunoblot analysis

Cells were lysed in ice-cold TNE buffer (50 mM Tris-HCl [pH 7.5], 150 mM NaCl, 1 mM EDTA) containing 1% NP-40, 1% TX-100 and protease inhibitors. For microsomal and cytoplasmic fractions of cultured cells, cell suspensions in ice-cold fractionation buffer (50 mM Tris-HCl, pH 7.5, 0.3 M sucrose, and protease inhibitors) were passed through a 26-gauge needle 10 times. The lysates were centrifuged at $8000 \times g$ for 10 min, and then the supernatant was further centrifuged at $100,000 \times g$ for 1 h. The resulting pellet was suspended with fractionation buffer with 0.2% NP-40, and was used as microsomal fraction. The supernatant was used as the cytoplasmic fraction. For purification of ufmylated CYB5R3, HEK293T cells expressing FLAG-6xHis-tagged CYB5R3 were lysed by TNE without EDTA, and the lysate was then centrifuged at $20,000 \times g$ for 10 min at 4 °C to remove debris. Thereafter, Ni-NTA agarose (R90101, Thermo Fisher Scientific) was added to the lysate, and the mixture was shaken under constant rotation for 30 min at 4 °C and then centrifuged at $1000 \times g$ for 2 min at 4 °C. The resulting precipitate was suspended with denaturing binding buffer (8 M urea, 20 mM sodium phosphate, pH 7.8, and 500 mM NaCl), and the mixture was rotated for 30 min at room temperature. The precipitates were washed three times with denaturing wash buffer (8 M urea, 20 mM sodium phosphate, pH 6.0, and 500 mM NaCl) and subsequently three times with wash buffer (20 mM sodium phosphate, pH 6.0, and 500 mM NaCl). To elute proteins, elution buffer (20 mM sodium phosphate, pH 4.0, 500 mM NaCl, and 500 mM imidazole) was added to the complex, and the mixture was centrifuged at $10,000 \times g$ for 10 min at room temperature. Samples were subjected to SDS-PAGE, then transferred to a polyvinylidene difluoride membrane (IPVH00010; Merck Millipore, Burlington, MA, USA). Antibodies against UFM1 (ab109305, Abcam, Cambridge, UK; 1:1000), UFSP2 (ab185965, Abcam; 1:1000), CYB5R3 (for human cell lines) (GTX84646; GeneTex, Irvine, CA, USA; 1:1000), CYB5R3 (for mouse cell lines) (10894-1-AP; Proteintech, Rosemont, IL, USA; 1:1000), UFBP1 (21445-1-AP, Proteintech; 1:1000), UFL1 (A303-456A; Bethyl Laboratories, Montgomery, TX, USA; 1:1000), CDK5RAP (H00080279-M01; Novus Biologicals, Englewood, CO, USA; 1:500), calnexin (sc-46669; Santa Cruz Biotechnology. Dallas, TX, USA; 1:500), GAPDH (MAB374, Merck Millipore; 1:1000), PRL26 (ab59567, Abcam; 1:1000), ACTIN (A1978; Sigma-Aldrich, Burlington, MO, USA; 1:2000), MYC (M192-3, Medical & Biological Laboratories, Nagoya, Japan; 1:1000) and FLAG (M185-3L, Medical & Biological Laboratories; 1:1000) were purchased from the indicated suppliers. Blots were incubated with horseradish peroxidase-conjugated goat anti-mouse IgG (H + L) (115-035-166, Jackson ImmunoResearch Laboratories, Inc.; 1:10000) or goat anti-rabbit IgG (H + L) (111-035-144, Jackson ImmunoResearch Laboratories, Inc.; 1:10000), and visualized by chemiluminescence. Band density was measured using the software Multi Gauge V3.2 (FUJIFILM Corporation, Tokyo, Japan). Uncropped and unprocessed scans of all immunoblots were supplied in the Source Data file.

## Immunoprecipitation

For immunoprecipitation analysis under non-denaturing conditions, cells were lysed in 300 µl of IP buffer (20 mM Tris-HCl [pH 7.5], 150 mM NaCl, 1 mM EDTA, 1% NP-40 and 1% TX-100) containing Protease Inhibitor Cocktail (Roche), and the lysates were then centrifuged at $20,000 \times g$ for 10 min at 4 °C to remove debris. In the next step, 200 µl of IP buffer, 2 µl of anti-CYB5R3 antibody (GTX84646, GeneTex), 1 µl of

anti-UFL1 antibody (A303-456A, Bethyl Laboratories, Inc.), and either 10 µl of Protein G Sepharose 4 Fast Flow (Cytiva) or 10 µl of anti-FLAG M2 Affinity Agarose Gel (A2220, Merck Millipore) were added to the 200 µl of lysate, and the mixture was mixed under constant rotation for 3 h at 4 °C. The immunoprecipitates were washed five times with ice-cold IP buffer. The complex was boiled for 5 min in SDS sample buffer in the presence of β-mercaptoethanol to elute proteins. For immunoprecipitation analysis of cytoplasmic and microsomal fractions under denaturing conditions, 10 µl of 10% SDS was added to 100 µl of both fractions. In the next step, 900 µl of IP buffer, 2 µl of anti-CYB5R3 antibody (GTX84646, GeneTex), and 10 µl of Protein G Sepharose 4 Fast Flow (Cytiva, Marlborough, MA, USA) were added to the 100 µl of cytoplasmic and microsomal fractions, and the mixture was mixed under constant rotation for 3 h at 4 °C. The immunoprecipitates were washed five times with ice-cold IP buffer. The complex was boiled for 5 min in SDS sample buffer in the presence of β-mercaptoethanol to elute proteins.

## Pull-down assay

Recombinant GST-fused proteins and MBP-fused proteins were produced in Escherichia coli and purified by chromatography on glutathione-Sepharose 4B resin (Amersham Biosciences, Arlington Heights, IL, USA) and amylose resin (New England Biolabs, Ipswich, MA, USA), respectively. The purified proteins were mixed in TNE buffer for 1 h at 4 °C before precipitation with glutathione-Sepharose. The mixtures were washed five times with ice-cold TNE. The bound proteins were analyzed by SDS-PAGE followed by Coomassie brilliant blue (CBB) staining or immunoblot analysis.

## Immunofluorescence microscopy

Cells grown on coverslips were fixed in 4% paraformaldehyde in PBS for 10 min, permeabilized with 0.1% Triton-X-100 or 50 ng/ml Digitonin in PBS for 5 min, blocked with 0.1% (w/v) gelatin (Sigma-Aldrich) in PBS for 45 min and then incubated overnight with primary antibodies diluted 1:200 in gelatin/PBS. Antibodies against MYC (M192-3, Medical & Biological Laboratories; 1:200), KDEL (ADI-SPA-827-D, Enzo Life Sciences, Farmingdale, NY, USA, 1:200), PDI (sc-20132, Santa Cruz Biotechnology, 1:200), FIP200 (17250-1-AP, Proteintech, 1:200),WIPI2 (ab105459, Abcam; 1:200), LC3 (PM036, Medical & Biological Laboratories; 1:200) and GABARAP (PM037, Medical & Biological Laboratories; 1:200) were purchased from the indicated suppliers. After washing, cells were incubated with Goat Anti-Rabbit IgG (H + L) Cross-Adsorbed Secondary Antibody, Alexa Fluor 568 (A11036, Thermo Fisher Scientific; 1:1000), and Goat Anti-Mouse IgG (H + L) Highly Cross-Adsorbed Secondary Antibody, Alexa Fluor 647 (A21236, Thermo Fisher Scientific; 1:1000) at a dilution ratio of 1:1000 for 60 min. Cells were imaged using a confocal laser-scanning microscope (FV1000, Olympus) with a UPlanSApo ×60 NA 1.40 oil-immersion objective lens. After image acquisition, contrast and brightness were adjusted using Photoshop ver 22.5.9 (Adobe). For the ER-phagy assay, HeLa cells expressing the ER-phagy reporter CYB5R3-mCherry-GFP were cultured in DMEM containing 10% FBS, 2 mM L-glutamine, and 5 U/ml penicillin or amino acid-free DMEM followed by confocal microscopy analysis. The number of RFP-positive punctae per cell was counted using a Benchtop High-Content Analysis System (CQ1, Yokogawa Electric Corp., Tokyo, Japan) and CellPathfinder software ver 3.06.01.08 (Yokogawa Electric Corp.).

## Size measurements and histological analysis of mouse brain

Mice were fixed by cardiac perfusion with 0.1 M phosphate buffer (PB, pH 7.4) containing 4% paraformaldehyde and 4% sucrose. Each brain was carefully dissected for measurements of the transverse and longitudinal lengths, then processed for paraffin embedding. Sections were prepared for hematoxylin and eosin staining, and images were captured with a BX51 microscope (Olympus, Tokyo, Japan).

## In vitro conjugation assays

GST-UFM1ΔC2, GST-UBA5, GST-UFC1, and GST-CYB5R3ΔN26 were transformed into *E. coli* Rosetta (DE3). The *E. coli* were cultured at 37 °C until OD600 reached 0.45, and then isopropyl-1-thio-β-D-galactopyranoside (final 250 mM) was added to induce the expression of each recombinant protein. The *E. coli* were further cultured at 25 °C for 5 h and collected by centrifugation at 2000 × g for 10 min at 4 °C. The pellets were lysed with lysis buffer (50 mM Tris-HCl (pH 8.5), 150 mM NaCl, 1% Triton-X, 1% NP-40, 1 mM dithiothreitol (DTT)) and then sonicated, and the lysates were centrifuged at 20,000 × g at 4 °C for 10 min. The resultant supernatant was mixed with Glutathione-Sepharose 4B resin (Amersham Biosciences). After washing, the GST tag was cleaved with PreScission Protease (27084301; Cytiva) to elute each recombinant protein. FLAG-UFL1 and UFBP1-MYC were co-expressed in *UFC1*-deficient HEK293T cells and the FLAG-UFL1 and UFBP1-MYC complex was purified by immunoprecipitation with Anti-DDDDK-tag pAb-Agarose (PM020-8; Medical & Biological Laboratories). Purified recombinant proteins, specifically UFM1ΔC2, UBA5, UFC1, FLAG-UFL1-UFBP1-MYC complex, and CYB5R3ΔN26 were dialyzed in 50 mM Tris (pH 8.5), 150 mM NaCl and 1 mM DTT (reaction buffer). UFM1ΔC2 (0.5 μM), UBA5 (0.5 μM), UFC1 (1 μM), and the UFL1-UFBP1 complex (0.1 μM) were mixed with a microsomal fraction (4 μg) prepared from *UFBP1*-deficient HEK293T cells or with CYB5R3ΔN26 (1 μM) in 20 μL of a reaction buffer containing 2 mM ATP and 10 mM MgCl₂. The mixtures were incubated at 30 °C for 90 min and the reaction was stopped by the addition of SDS sample buffer containing 5% β-mercaptoethanol.

The expression of GST-CYB5R3ΔN26 in *E. coli* Rosetta (DE3) was induced as described above. To produce a large amount of ufmylated CYB5R3, purified Strep-UFM1ΔC2 (9 μM), UBA5 (2 μM), UFC1 (5 μM), and lysates of *E. coli* Rosetta (DE3) expressing GST-CYB5R3ΔN26 (5 mg/ml) were mixed in 50 ml of a reaction buffer containing 2 mM ATP and 10 mM MgCl₂. The mixtures were incubated at 30 °C for 90 min. GST-CYB5R3ΔN26 conjugated with Strep-UFM1 was purified with Glutathione-Sepharose 4B resin (Amersham Biosciences), and the GST-tag was cleaved using PreScission Protease (27084301; Cytiva). Thereafter, UFM1-conjugated CYB5R3ΔN26 was purified with Strep-Tactin Sepharose (IBA Lifesciences, Gottingen, Germany), and free GST-CYB5R3ΔN26 was removed.

## X-ray crystal structural analysis of CYB5R3

Purified CYB5R3 for the in vitro conjugation assay was further subjected to size-exclusion chromatography with 20 mM HEPES (pH 7.4) and 150 mM sodium chloride using a Superdex 200 10/300 column (GE Healthcare) and used for crystallization after concentration. Crystallization was performed by the sitting-drop vapor diffusion method at 20 °C. CYB5R3 at 17 mg/ml was mixed with an equal amount of reservoir solution consisting of 30% PEG1500, 10% 2-propanol, and 0.1 M bicine at pH 8.5 and equilibrated against the reservoir solution for 1 week. Crystals were frozen in liquid nitrogen and kept in a stream of nitrogen gas at −178 °C during diffraction data collection, which was performed using a beamline BL32XU at SPring-8 in Japan, with a wavelength of 1.0000 Å. The diffraction data were indexed, integrated, and scaled using XDS (ver. May 1, 2016)[51]. Structural determination was performed by the molecular replacement method using Phenix (ver. 1.11.1)[52] software. The crystal structure of pig CYB5R3 (PDB ID 3W5H) was used as a search model. Crystallographic refinement was performed using Phenix and COOT (ver. 0.8.9)[53] software. The parameters of diffraction data collection and refinement are summarized in Table 1.

## X-ray crystal structural analysis of UFBP1(UFIM)-UFM1 fusion

For structural determination of UFM1 complexed with the UFIM of UFBP1, we used the inverse PCR method with pGEX6p-Strep-UFM1 as a template to construct a pGEX6p-UFBP1 (residues 194–202)-UFM1 vector that encoded GST-tagged UFBP1 UFIM

directly fused at the N-terminus of UFM1. GST-UFBP1 UFIM-UFM1 was expressed in *E. coli* BL21(DE3) cells. After purification with GST-Accept resin (Nacalai Tesque), GST was cleaved from the protein using an HRV 3 C protease. After the buffer was exchanged with PBS, the protein was again applied to GST-Accept resin and the flow-through fraction was collected. Finally, the protein was analyzed with Superdex 200 26/600 size-exclusion chromatography (GE Healthcare) and eluted with 20 mM Tris-HCl (pH 8.0) and 150 mM NaCl. Crystallization was performed by the sitting-drop vapor diffusion method at 20 °C. UFBP1 UFIM-UFM1 fusion at 44 mg/ml was mixed with an equal amount of the reservoir solution consisting of 28% 2-propanol, 3% PEG 200, and 0.1 M MES at pH 6.0 and equilibrated against the reservoir solution for a few days. Crystals were soaked with reservoir solution supplemented with 20% glycerol, then frozen and kept in a stream of nitrogen gas at −180 °C during diffraction data collection. Diffraction data collection was performed using XtaLAB Synergy Custom (Rigaku), an in-house rotating anode X-ray generator equipped with a HyPix-6000HE hybrid photon-counting X-ray detector with a wavelength of 1.5418 Å. The diffraction data were indexed and integrated using a CrysAlis Pro (Rigaku), and scaled using Aimless in the CCP4 package[54]. Structure determination was performed by the molecular replacement method using MolRep ver 11.0[55]. The crystal structure of UFM1 (PDB ID 5HKH) was used as a search model. Crystallographic refinement was performed using Phenix and COOT software. The parameters of diffraction data collection and refinement are summarized in Table 1.

## HS-AFM imaging of CYB5R3ΔN26

HS-AFM images were acquired in tapping mode using a sample-scanning HS-AFM instrument (MS-NEX; Research Institute of Biomolecule Metrology, Tsukuba, Ibaraki, Japan). CYB5R3ΔN26 (~100 nM) in 2 μL of observation buffer (20 mM HEPES-NaOH (pH 7.0) containing 150 mM NaCl) was deposited onto a freshly cleaved mica substrate attached to the top of a glass stage (diameter, 1.5 mm; height, 2 mm). After 3-min incubation, the mica surface was rinsed with an observation buffer. We used cantilevers measuring ~7 μm long, ~2 μm wide, and ~0.08 μm thick with electron beam–deposited (EBD) tips (tip radius <10 nm) (USC-F1.2-k0.15; NanoWorld). Their resonant frequency and spring constant were 1.2 MHz in air and 0.15 N/m, respectively. Imaging conditions were as follows: scan size, 60 nm × 60 nm; pixel size, 100 × 100 pixels; imaging rate, 5 frames/s. Imaging was performed at ~23 °C. HS-AFM images were viewed and analyzed using Kodec software ver 4.4.7.39[56].

## HS-AFM data analysis

HS-AFM images were viewed and analyzed using Kodec software[56]. Because the molecules sometimes change their orientation on the mica surface, we analyzed the images in which the two lobes (FAD and NADH domains) were facing up. The distance between the two lobes in each image was manually measured using the cross-section tool of Kodec. The histograms and the fitting curves with double-Gaussian distribution were produced using Origin software.

## Simulation of AFM images

The simulated AFM images shown in Fig. 2 were obtained by Biomolecular AFM viewer-2.1 software[57], using the X-ray crystal structure of CYB5R3 (PDB ID: 1UMK) and a modified form in which the hinge region (Leu147) was arranged. The AFM parameters were set as follows: scan step, 0.6 nm; cone angle, 20°; tip radius, 0.8 nm.

## Measurement of reductase activity

Microsomal fractions (0.5 mg), purified CYB5R3 (1 ng), or ufmylated CYB5R3 (1 ng) were incubated with 10 mM Tris-HCl (pH 7.5) containing

2 mM $K_3Fe(CN)_6$ and 250 µM NADH in a final volume of 200 µl for 10 min. The reduction rate by CYB5R3 at 420 nm was assessed via spectrophotometry.

## Mass spectrometry analysis

Pulled-down proteins were separated by SDS-PAGE, then the bands were excised and digested with trypsin. The tryptic peptide samples were separated using a nanoLC Ultra 1D plus nano liquid chromatography system (SCIEX, Eksigent Technologies; ChromXP; 150 µm × 10 cm, 3 µm particle), by a 60 min gradient from 0–30% solvent B (solvent A: 0.1% formic acid, solvent B: 0.1% formic acid in acetonitrile) at a flow rate of 300 nl/min. Eluted peptides were directly analyzed by a 5600 Triple TOF mass spectrometer (SCIEX) that was operated in data-dependent acquisition mode. Analysit TF® software (ver.1.6.0, Sciex) was used for data acquisition and processing. The raw mass spectrometric files were analyzed with ProteinPilot version 5.0.1 software (SCIEX) using the Paragon™ algorithm (5.0.0.0. 4767) against the uniport 2015.03 database (40404 entries) supplemented with 245 frequently observed contaminants, including human keratins, bovine serum proteins, and proteases. The following search parameters were applied: cys. alkylation, none; enzyme, trypsin; special factors, gel-based ID and UFMylation; species, homo sapiens: ID focus, biological modifications; search effort, thorough; detected protein threshold [Conf] >10%; automatic false discovery rate (FDR) analysis. Protein identifications were further filtered at a level of 1% FDR. Among the proteins identified with >99% confidence and minimum 2 unique peptides, those exclusively found in HEK293T cells expressing FLAG-His-UFM1ΔC2 but not FLAG-His-UFM1ΔC3 were analyzed as UFM1-interacting proteins. Based on this criterion, we reasoned that 60 of 151 proteins originally identified in the co-immunoprecipitation assay using FLAG-His-UFM1ΔC2 were UFM1-interacting proteins (Supplementary Data 1). A complete list of the proteins and peptides identified in the experiment is attached (Supplementary Data 1). The mass spectrometry proteomics data have been deposited to the ProteomeXchange Consortium (http://proteomecentral.proteomexchange.org) via jPOSTrepo[58] (PXD021225).

To identify UFM1-binding proteins under expression of UFL1 and UFBP1, FLAG-UFM1 alone or in various combinations with MYC-UFL1 and UFBP1-MYC was expressed in HEK293T cells. As a control, FLAG vector (mock) was also transfected in HEK293T cells. The cell lysates were subjected the immunoprecipitates to mass spectrometry (MS) ($n = 4$)[32]. Digested peptide samples were analyzed using a nanoscale LC-MS/MS system[59]. The peptide mixture was applied to a Mightysil-PR-18 (Kanto Chemical) frit-less column (45 × 0.150 mm ID) and separated using a 0–40% gradient of acetonitrile containing 0.1% formic acid for 80 min at a flow rate of 100 nL/min. Eluted peptides were sprayed directly into a QSTAR Elite mass spectrometer (SCIEX). MS and MS/MS spectra were obtained using the information-dependent mode. MS and MS/MS mass spectra were recorded in positive ion mode with a resolution of 12,000–15,000 full-width half-maximum. MS mass range was m/z 350–1500 and MS/MS mass range was m/z 100–1500. The precursor ions were fragmented in a collision cell using nitrogen as the collision gas. MS/MS spectra were searched against the database using precursor ion tolerance values ranging from the 250 p.p.m. range to 0.25 Da. Protein identification was performed with the Mascot Server v.2.3 search engine (Matrix Science, Boston, MA, USA) using the NCBI nonredundant human protein data set (NCBInr RefSeq Release 71, containing 179460 entries). Protein quantification was performed using the iBAQ method[60] without conversion to absolute amounts using universal proteomics standards. The data were searched with the following modifications: carbamidomethyl as variable modifications. A maximum of 2 missed cleavages was allowed. Minimum peptide length is 7. The iBAQ value was calculated by dividing the sum of the ion intensities of all the identified peptides of each protein by the number of theoretically measurable peptides. Among the identified proteins,

we extracted those that were not identified in mock MS and those that were identified in mock MS but whose iBAQ values were less than one-fifth of the other samples. The candidate proteins were further narrowed down by selecting those that were detected more than once only when both MYC-UFL1 and UFBP1-MYC were expressed. The complete list of proteins identified in the experiment is attached (Supplementary Data 2). The mass spectrometry proteomics data have been deposited to the ProteomeXchange Consortium (http://proteomecentral.proteomexchange.org) via jPOSTrepo[58] (PXD038409).

## Lipidome analysis

Lipids were extracted by the method of Bligh and Dyer with internal standards. The organic (lower) phase was transferred to a clean vial and dried under a stream of nitrogen. The lipids were then resolubilized in methanol, and a portion of the extracted lipid was injected onto the liquid chromatography/tandem mass spectrometry (LC-MS/MS) system. LC separation was performed on an ACQUITY UPLC™ BEH $C_{18}$ column (1.7 µm, 2.1 × 100 mm; Waters, Milford, MA, USA) coupled to an ACQUITY UPLC™ BEH $C_{18}$ VanGuard™ Pre-column (1.7 µm, 2.1 × 5 mm; Waters). Mobile phase A was 60:40 (v/v%) acetonitrile/$H_2O$ containing 10 mM ammonium formate and 0.1% (v/v) formic acid, and mobile phase B was 90:10 (v/v%) isopropanol/acetonitrile containing 10 mM ammonium formate and 0.1% (v/v) formic acid. The LC gradient consisted of 20% B for 2 min, a linear gradient to 60% B over 4 min, a linear gradient to 100% B over 16 min and equilibration with 20% B for 5 min (27 min total run time). The flow rate was 0.3 mL/min, and the column temperature was 55 °C. Multiple reaction monitoring was performed using a Xevo™ TQ-S micro triple quadrupole mass spectrometry system (Waters) equipped with an electrospray ionization (ESI) source. The ESI capillary voltage was set at 1.0 kV, and the sampling cone was set at 30 V. The source temperature was set at 150 °C, desolvation temperature was set at 500 °C and desolvation gas flow was 1000 L/h. The cone gas flow was set at 50 L/h.

## Statistical analysis

Values, including those displayed in the graphs, are means ± s.e. Statistical analysis was performed using the unpaired t-test (Welch test) or Šidák's multiple comparison test by GraphPad Prism ver 9.2.0 software (GraphPad software). A $P$ value <0.05 was considered to indicate statistical significance.

## Reporting summary

Further information on research design is available in the Nature Portfolio Reporting Summary linked to this article.

# Data availability

The proteomics data generated in this study have been deposited in the ProteomeXchange under accession code PXD021225 and PXD038409. Coordinates and structure factors of UFBP1 UFIM-UFM1 fusion and CYB5R3 have been deposited in the Protein Data Bank under accession codes 7W3N and 7W3O, respectively. All figures and movies are available in figshare [https://doi.org/10.6084/m9.figshare.21641051]. Source data are provided with this paper.

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

## Acknowledgements

We thank Yuki Ishii for assistance with crystallization. R.I. is supported by a Grant-in-Aid for Scientific Research (C) (22K06931). N.N.N. is supported by a Grant-in-Aid for Scientific Research on Innovative Areas (19H05707), JST CREST (JPMJCR20E3), and by the Platform Project for Supporting Drug Discovery and Life Science Research (Basis for Supporting Innovative Drug Discovery and Life Science Research (BINDS)) from AMED under grant no. JP19am0101001 (support no. 0002). M.K. is supported by a Grant-in-Aid for Scientific Research on Innovative Areas (19H05706), a Grant-in-Aid for Scientific Research (A) (21H004771), Advanced Research and Development Programs for Medical Innovation (AMED-CREST, 22gm1410004s0103), the Japan Society for the Promotion of Science (an A3 foresight program) and the Takeda Science Foundation (to M.K.). This work was supported by JSPS KAKENHI Grant Number JP 16H06276, JP 22H0492 (AdAMS). A.H.E-.G. would like to thank the Egyptian Government for the financial support of this research project.

## Author contributions

R.I., N.N.N., and M.K. designed and directed the study. R.I., A.H.E.-G., and T.U. carried out the biochemical and cell biological experiments. S.K.-H. generated knockout cell lines. M.S., Y.O., T.H., T.N., and H.S. conducted proteomic analyses. H-C.L. and T.Y. performed lipidomics. D.N. and N.N.N. performed high-speed AFM analysis. M.A. and K.S. generated genetically modified mice. T.U. and S.W. carried out the histological analysis of genetically modified mice. N.N.N. performed X-ray structural analysis. T.I. and TM.M. provided intellectual support. M.K. and N.N.N. wrote the manuscript. All authors discussed the results and commented on the manuscript.

## Competing interests

The authors declare no competing interests.

## Additional information

¹Department of Physiology, Juntendo University Graduate School of Medicine, Bunkyo-ku, Tokyo 113-8421, Japan. ²Biochemistry Division, Chemistry Department, Faculty of Science, Tanta University, Tanta 31527, Egypt. ³Division of Biological Molecular Mechanisms, Institute for Genetic Medicine, Hokkaido University, Sapporo 060-0815, Japan. ⁴Calpain Project, Department of Basic Medical Sciences, Tokyo Metropolitan Institute of Medical Science, Setagaya-ku, Tokyo 156-8506, Japan. ⁵Advanced Technical Support Department, Center for Basic Technology Research, Tokyo Metropolitan Institute of Medical Science, Setagaya-ku, Tokyo 156-8506, Japan. ⁶National Institutes of Advanced Industrial Science and Technology, Biological Information Research Center (JBIRC), Kohtoh-ku, Tokyo 135-0064, Japan. ⁷Department of Animal Model Development, Brain Research Institute, Niigata University, Chuo-ku, Niigata 951-8585, Japan. ⁸Department of Anatomy and Histology, Fukushima Medical University School of Medicine, Hikarigaoka, Fukushima 960-1295, Japan. ⁹Department of Biochemistry, Juntendo University Graduate School of Medicine, Bunkyo-ku, Tokyo 113-8421, Japan. ¹⁰Laboratory of Proteomics and Biomolecular Science, Biomedical Research Core Facilities, Juntendo University Graduate School of Medicine, Bunkyo-ku, Tokyo 113-8421, Japan. ¹¹Division of RNA and gene regulation, Institute of Medical Science, The University of Tokyo, Minato-Ku 108-8639, Japan. ✉e-mail: mkomatsu@juntendo.ac.jp

