## [Peer Review File · Nature Communications]

The UFM1 system regulates ER-phagy through the ufmylation of CYB5R3Editorial Note: This manuscript has been previously reviewed at another journal that is not operating a transparent peer review scheme. This document only contains reviewer comments and rebuttal letters for versions considered at *Nature Communications*.

REVIEWER COMMENTS

Reviewer #1 (Remarks to the Author):

In this revised manuscript the authors have taken steps to address the deficiencies present in the original manuscript. The principal conclusion of the work – that CYB5R3 is a bona fide endogenous substrate of UFMylation is still unconvincing, however. The data show as in the previous version, that CYB5R3 -His-FLAG, when massively overexpressed in HEK293T cells together with UFM1 and two components of the UFM1 E3 (UFBP1 and UFL1) can be modified with a single UFM1 moiety on a single lysine residue K214.

1. The revised manuscript now includes one experiment on endogenous (ie not overexpressed) CYB5R3 UFMylation (Fig 2a) (where the UFMylation components are not overexpressed) showing a very faint band putatively identified on a CYB5R3 western blot in lysates and enriched in CYB5R3 pulldowns only upon overexpression of the E3 components UFL1 and UFBP1. The authors' prime argument supporting the conclusion that CYB5R3 is a bona fide substrate rests on the data in Fig 2a from lane 5 where they IP CYB5R3 and claim to see the UFM1- CYB5R3 adduct in cells not overexpressing UFL1/UFBP1. However, the putative adduct actually does not appear to have the same mobility as the adducts in the E3 overexpressing lanes (compare Fig 2a lane 5 with lanes 3 and 7 in the anti CYB5R3 panel). In Fig 2b they show a small (<50%) reduction in the intensity of the putative endogenous CYB5R3 adduct following siRNA to UFBP1 when blotted for CYB5R3 but complete loss of what they argue to be the same band in the UFM1 blot. Since the same exact samples are being analyzed in the two blots, it is hard to understand the discrepancy. Perhaps they're not the same bands?

2. Remarkably, the same lysine specificity on CYB5R3 can be recapitulated in a cell free reaction with only the E1 and E2. In other words when there is enough CYB5R3 and charged E2 present selective UFMylation of a single lysine does not depend on a the E3-the specificity determining component in all UBL systems! This suggests that K214 is selectively modified by UFM1 because it is a better nucleophile than the other lysines and argues against UFMylation of CYB5R3 being a physiologically meaningful process.

3. The fact that UFSP2 KO increases the abundance of this species does not substantiate the claim of specificity because UFSP2, recognizes the UFM1 moiety and will cleave all UFM1 adducts.

4. The observation that UFM1-modified CYB5R3 is enriched in membrane fractions does not contribute to substantiating the authenticity of CYB5R3 being a substrate as both it and the UFM1 conjugation machinery are present at the same ER membrane Independently of one another. Moreover , the

fractionation is problematic. Cytosol fractions contain ~50% contamination with ER based on BiP distribution in some figures (eg Fig 2)

5. Throughout the manuscript the authors quantify the ratio of UFMylated to unmodified CYB5R3 in bar graphs and note extremely low P-values supporting their conclusions, but it is unclear how these ratios are calculated. In most cases, the fraction of CYB5R3 that's modified is well below 10% (difficult to accurately assess because the unmodified form is so overexposed) yet their quantification shows ratios that must be normalized in some way that is not explained and not consistent. For example compare quantification of UFM1 modified CYB5R3 in Fig 2 with quantification of GFP-UFM1 modified CYB5R3 in Fig 5. Every figure in the paper has at least one panel using a similar quantification approach where the quantification does not seem to correspond to the actual data in the western blots.

6. In Fig 4 the authors use a pH-quenched fluorescent reporter attempt to demonstrate a role for UFMylation in promoting lysosomal degradation of CYB5R3. The visual data presented in panels b and c are impossible to interpret by eye so the authors quantify red puncta and use a statistical approach to claim a significant increase upon overexpression of UFL1/UFBP1 (and of course, the substrate cherry/GFP- CYB5R3) in starved cells. It is unclear why starvation was needed. Visual inspection of the graphical data show large error bars and many outlier points that are difficult to reconcile with the reported P-values, but I am not a statistician.

7. The authors have -somewhat arbitrarily decided to consider the third component of the tripartite UFM1 E3, CDK5RAP3, as a "lysosome autophagy adaptor" based on data in a single report, while ignoring a substantial body of data supporting the view that this protein is a core constituent of the E3. Indeed, functional insight into the role of this E3 component was recently published on BioRxiv. While it can't be ruled-out that CDK5RAP3 could have two functions, it is concerning that its function in UFM1 ligase activity was not considered in the experimental design or in the data interpretation. It is difficult to understand their rationale for performing all the reconstitution and overexpression experiment with only two of the three components of this E3 (and indeed they show, disconcertingly, that the E3 is entirely dispensable for CYB5R3 UFMylation in vitro). In Fig 5a they finally test the role for CDK5RAP3 and find that its inclusion in the overexpression paradigm actually reduces the amount of UFMylated CYB5R3 (and UFBP1). While one explanation for this result is that CDK5RAP3 promotes the specificity of the E3 for its authentic substrate (in part by suppressing modification of non-authentic substrates) they choose to interpret the CDK5RAP3 effect as supporting a role in lysosomal degradation. But the experiment they present in support of this hypothesis, in Fig 5b (anti CYB5R3 blot) shows a remarkably weak effect of the lysosomal inhibitor BafA (compare lanes 5 and 6). Although at best there is a 20% increase in the GFP-UFM1- CYB5R3 band in the blot, strangely the quantification reports a highly significant ($P = 0.003$) effect.

8.

9. The structure/model in Fig 2g claims to show a steric clash that is not at all clear from the figure.

10. The claim that CYB5R3 reductase activity is lost from Fig 2i is not convincing. How are the results normalized? Do they have equivalent amounts of protein? What if the mutation doesn't bind FAD as tightly?

Reviewer #3 (Remarks to the Author):

All comments and concerns raised by the reviewer have been adequately addressed by the authors. I recommend publication of the revised manuscript.

Reviewer #4 (Remarks to the Author):

There is no doubt that the manuscript is greatly improved and that the majority of concerns of the original reviewers have been addressed but a few points should be considered when making the final decision:

Reviewer 1's comments are largely addressed adequately bar the following minor point:

- I think confirmation that the new endogenous CYB5R3 from microsomal fractions and probing for UFL1 is done under denaturing conditions would help. The methods suggest this is the case but I would make it more explicit in the text and figure legend. If it is not, it should be.

Reviewer 2's comments are largely addressed adequately bar the following minor points:

- there are still limitations on data showing the endogenous nature of some of the molecular details of the mechanism. Immunofluorescence of UFL1 localisation remains absent, but on balance it may be impossible to achieve this with current reagents. Short of knock-in of a tag to the endogenous locus using CRISPR/Cas9, which would be extremely time consuming and has no guarantee of methodological success, I think there is little else on balance that can be done here. Overall, although this point has not been addressed completely, I don't think it should preclude publication at this stage and in this journal.

- similarly, there is still a reliance on ectopic expression when using the Fluoppi system to show ATG8 protein interaction with CDK5RAP3. Here I think the authors should at least provide additional data to show that the expression level of these proteins is comparable with endogenous and that mutation of known binding interfaces on ATG8 proteins, e.g. the LDS (if these are indeed involved in CDK5RAP3 interaction) can reduce the Fluoppi signal, thus validating the specificity of this system for detecting ATG8-adaptor/receptor interactions.

All reviewer 3's comments appear to be addressed adequately.

My additional comments - while I do not wish to add undue burden to the revision of this manuscript there is a fundamental point that is not specifically identified by any of the three prior reviewers but that I think is critical to draw the major conclusions of the manuscript. That is, proof of CYB5R3 turnover by autophagy is not per se proof of ER-phagy, despite the outer ER localisation of at least a substantial population of this protein. The authors should really use an abundant general ER marker and show degradation by ER-phagy occurs in an at least partially CYB5R3-dependent manner. For instance, ss-mRFP-GFP-KDEL, or tandem fluorescent RAMP4, etc., could be employed in a tandem fluorescent flux assay, analogously to how CYB5R3 is currently used. Similarly, the CYB5R3 positive autophagic foci identified by colocalisation with autophagy proteins should be demonstrated to contain other ER material via microscopy. N.B. I apologise if some of this was present in the previous version of the manuscript and has been removed, as I have not seen this draft. The response to Reviewer 1 might hint at this with regard to the mRFP-GFP-KDEL. I wasn't completely clear on this.

Reviewer #5 (Remarks to the Author):

The revised manuscript has been improved significantly. However, as the reviewer #1 stated: the principal conclusion of the work – that CYB5R3 is a bona fide endogenous substrate of UFMylation is still unconvincing. He also raised many other issues, which are reasonable.

Although almost all the experiments are done in the overexpressed system and the final in vivo experiments could not guarantee that the effect is caused by the ufmylation BUT NOT by other lysine modifications at the position 214, currently there are no better experiment techniques which could solve this problem.

There are still some minor issues in the manuscript, which need to be fixed before the manuscript is publishable. Because some issues require further experiments, therefore, I suggested a major revision of the manuscript. Please see below for other issues.

1. Figure 1a: The BiP distribution is different in the USFP2 KO cells? What is the reason for this? It should be discussed in the manuscript. Figure 1e: Why does the lower position have double bands? Figure 1f:

Should CYB5R3-GFP-UFM1 be written as GFP-UFM1-CYB5R3? Figure 1i: Should CYB5R3-Strep-UFM1 be written as Strep-UFM1-CYB5R3?

2. Figure 1d showed that UFL1 is important for the ufmylation of CYB5R3. However, in Figure 1f, ufmylation of CYB5R3 occurs in the absence of UFL1. How could this happen? What caused this discrepancy?

3. Figure 2a: Why is there no heavy chain but with intense light chain? Figure 2a and 2b: Add "Relative" in the y axis of the statistical data. Figure 2b: IP and WB are required to confirm the ufmylation of endogenous CYB5R3. It is hard to tell whether the bands are specific or not if no IP is performed to eliminate the effect from other interacting proteins. The current images showed that there is a slightly less intense band when UFL1 and UFBP1 is knocked down in the anti-CYB5R3 blot. However, in the anti-UFM1 WB, this band is completely disappeared. This inconsistent might indicate this band is not ufmylated CYB5R3. A stricter experiment should be provided to confirm this result. Figure 2c: Why is the position for CYB5R3-HF in the microsome and cytosol different in the top image of Figure 2c? Figure 2j: Why does not the KR mutation affect Vi in the absence of UFM1, UFL1, and UFBP1? This result indicates the effect is not physiologically relevant.

4. Figure 3b and 3c: Why are there three bands in the top panel of Figure 3b? Blot or coomassie stain for GST should be provided in Figure 3b and 3c. Figure 3k: Which image, middle top image, was used for the quantification? Please indicate in the figure legend. Figure 3g: How can the authors rule out the interaction between UFM1 and UFBP1 when the authors demonstrate that the ufmylated CYB5R3 interacts with UFBP1? Based on the left panel of Figure 3g, UFBP1 interacts with UFM1.

5. Figure 5e: Please include three different experimental conditions in the last bar graph.

6. Supplementary Figure S5C: What is the lowest band in Western blotting image?

7. Supplementary Figure S6: The fractionation is not perfect. GAPDH is also present in the microsomal fraction and is not consistent under different conditions. (It is not a big problem).

8. Supplementary Table 1: Please explain the terms in detail. What are the peptides (95%)? Are they unique peptides or peptide spectrum matches?

9. Supplementary Table 2 and Supplementary Table 3: Please pay attention to the significant digits.

10. Page 4, line 120: Why is UFL1 not required for CYB5R3 ufmylation?

11. In the Discussion, the authors conclude that 2) ufmylated CYB5R3 interacts with the E3 ligase component UFBP1 to facilitate further ufmylation of CYB5R3. However, there is no sufficient evidence to conclude this statement.

12. There are some typos for example, GFP-tagged UFL1 (GFP-UFL1) were...; bonafide

Reviewer #6 (Remarks to the Author):

My review of the manuscript is only based on the high-speed AFM part, given that this is my area of expertise. I will refrain from commenting on the much broader other aspect of the paper.

Appreciate the efforts that the authors have put in to investigating CYB5R3 with high-speed AFM to observe structural changes. Unfortunately very little information is given in the manuscript about the HS-AFM data, which makes it very difficult to interpret. Based on what criteria did the authors attribute the closed and open states of the molecule? What kind of differences do the authors expect based on the crystallography data? It seems to me that the fluctuations comparing images denoted as "O" with each-other are similar to the fluctuations comparing "O" with "C", or "C" with "C". HS-AFM experiments always have significant differences from experiment to experiment. How many different experiments have the authors performed? How many molecules have they observed per experiment? In how many cases did they see a convincing difference between "O" and "C"?

Reviewer #1:

General comment

In this revised manuscript the authors have taken steps to address the deficiencies present in the original manuscript. The principal conclusion of the work – that CYB5R3 is a bona fide
5 *endogenous substrate of UFMylation is still unconvincing, however. The data show as in the previous version, that CYB5R3 -His-FLAG, when massively overexpressed in HEK293T cells together with UFM1 and two components of the UFM1 E3 (UFBP1 and UFL1) can be modified with a single UFM1 moiety on a single lysine residue K214.*

10 Reply

Thank you for your valuable comments, which helped us revise and strengthen our original manuscript. According to the comments raised by Reviewers 1 and 5, we made the following changes.

1. To prove that CYB5R3 is a *bona fide* substrate for UFM1, we carefully re-performed
15 several experiments to detect endogenous ufmylated CYB5R3 and the knockdown-induced suppression of *UFL1* or *UFBP1* (Fig. 2a and b).
2. To demonstrate the requirement for E3 in CYB5R3 ufmylation, we performed an *in vitro* ufmylation assay (Fig. 1g).
3. We delineated the quantification and statistical methods in this study.
- 20 4. In the revised manuscript, we discussed the role of CDK5RAP3 as an E3 regulatory subunit with reference to related studies.

For specific details, please see our reply to each comment below.

Comment 1

- 25 *The revised manuscript now includes one experiment on endogenous (ie not overexpressed) CYB5R3 UFMylation (Fig 2a) (where the UFMylation components are not overexpressed) showing a very faint band putatively identified on a CYB5R3 western blot in lysates and enriched in CYB5R3 pulldowns only upon overexpression of the E3 components UFL1 and UFBP1. The authors' prime argument supporting the conclusion that CYB5R3 is a bona fide*
30 *substrate rests on the data in Fig 2a from lane 5 where they IP CYB5R3 and claim to see the UFM1- CYB5R3 adduct in cells not overexpressing UFL1/UFBP1. However, the putative adduct actually does not appear to have the same mobility as the adducts in the E3 overexpressing lanes (compare Fig 2a lane 5 with lanes 3 and 7 in the anti CYB5R3 panel). In Fig 2b they show a small (<50%) reduction in the intensity of the putative endogenous CYB5R3*

35 *adduct following siRNA to UFBP1 when blotted for CYB5R3 but complete loss of what they*
argue to be the same band in the UFM1 blot. Since the same exact samples are being analyzed
in the two blots, it is hard to understand the discrepancy. Perhaps they're not the same bands?

Reply 1

40 Thank you for pointing this out. We regret that the results presented in the original manuscript
were unclear. We agree with the reviewer that this experiment is key to assuring that CYB5R3
is an authentic substrate of UFM1. Therefore, we thoroughly optimized the experimental
settings and performed the experiment multiple times. As shown in Fig. 2a of the revised
manuscript, immunoprecipitation of microsomal fractions with CYB5R3 antibody revealed
45 endogenous ufmylated CYB5R3, and the same migrated band was significantly increased by
overexpression of E3 (UFL1 and UFBP1). Furthermore, as shown in Fig. 2b of the revised
manuscript, we showed that knockdown of *UFL1* or *UFBP1* significantly decreased the level of
endogenous ufmylated CYB5R3.

50 *Comment 2*

Remarkably, the same lysine specificity on CYB5R3 can be recapitulated in a cell free reaction
with only the E1 and E2. In other words when there is enough CYB5R3 and charged E2 present
selective UFMylation of a single lysine does not depend on a the E3-the specificity determining
component in all UBL systems! This suggests that K214 is selectively modified by UFM1
55 *because it is a better nucleophile than the other lysines and argues against UFMylation of*
CYB5R3 being a physiologically meaningful process.

Reply 2

This *in vitro* reaction is a very specific experimental system, *i.e.*, an artificial system for producing
60 large amounts of ufmylated CYB5R3. First, it used an excess quantity of recombinant UFM1 (9
 μM), UBA5 (2 μM), and UFC1 (5 μM). Second, instead of purified CYB5R3 $\Delta\text{N}26$, a lysate of *E.*
coli expressing GST-CYB5R3 $\Delta\text{N}26$, was mixed with these recombinant enzymes. Under these
conditions, CYB5R3 was covalently bound to UFM1 without E3. We cannot explain the exact
mechanism involved, so we omitted this data from Figure 1. However, CYB5R3 covalently bound
65 to UFM1 is a valuable tool, so we used it for the pull-down experiments (Fig. 3b and g).

As pointed out by Reviewers 1 and 5, we agree that in the original manuscript there
were problems in demonstrating the specificity of E3. To solve this issue, we expressed FLAG-
UFL1 and UFBP1-MYC in *UFC1*-deficient HEK293T cells and purified the FLAG-UFL1 and

UFBP1-MYC complex by immunoprecipitation with anti-FLAG antibody (anti-DDDDK-tagged pAb-agarose). *In vitro* ufmylation assays were performed using recombinant UFM1, UBA5, UFC1, the purified E3 complex, and a microsomal fraction prepared from *UFBP1*-deficient HEK293T cells or recombinant CYB5R3ΔN26 from *E. coli*. As shown in Fig. 1g of the revised manuscript, ufmylated CYB5R3 was not formed by the addition of UFM1, UBA5, and UFC1 to microsomal fractions, but it was formed by adding them together with purified E3 complex. By contrast, no ufmylated CYB5R3 was formed in the assay with recombinant CYB5R3ΔN26 (Supplementary Fig. S3c). This implies that E3 and one or more factors in the microsomal fractions are required for ufmylation of CYB5R3. We presented the aforementioned points in the revised manuscript.

80 *Comment 3*

The fact that UFSP2 KO increases the abundance of this species does not substantiate the claim of specificity because UFSP2, recognizes the UFM1 moiety and will cleave all UFM1 adducts.

Reply 3

85 The comment may be correct. However, it has been shown that RPL26, which is considered to be an authentic UFM1 substrate, is also increased by ablation or knockdown of *UFSP2* (Walczak et al., PNAS, 116:1299–1308, 2019; Wang et al., 30(1):5–20, Cell Res, 2019). Therefore, we believe that the data showing that *UFSP2* knockout increases ufmylated CYB5R3, together with Figs. 2a, 2b (detection of endogenous ufmylated CYB5R3 and inhibition of its formation by knockdown of E3 components), and 1f (requirement for E3 in the *in vitro* ufmylation assay) in the revised manuscript, indicate that CYB5R3 is a substrate of UFM1.

Comment 4

95 *The observation that UFM1-modified CYB5R3 is enriched in membrane fractions does not contribute to substantiating the authenticity of CYB5R3 being a substrate as both it and the UFM1 conjugation machinery are present at the same ER membrane Independently of one another. Moreover, the fractionation is problematic. Cytosol fractions contain ~50% contamination with ER based on distribution in some figures (eg Fig 2).*

100

Reply 4

As the reviewer pointed out, this data is only supportive evidence. However, now that the key issues of Fig. 2a and 2b (detection of endogenous ufmylated CYB5R3 and inhibition of its formation by knockdown of E3 components) and Fig. 1f (requirement for E3 *in vitro* ufmylation assay) are resolved, we would like to keep this data in the revised manuscript as well. In confirming our fractionation experiments, we believe that the Bip antibody we used in the blots was problematic. In fact, blots with anti-calnexin antibody (Santa Cruz Biotechnology, sc-46669), which has been used in many papers (e.g., Rajagopalan et al., Science, 263:387–90, 1994), showed clearer fractionation in comparison with the Bip antibody.

Comment 5

Throughout the manuscript the authors quantify the ratio of UFMylated to unmodified CYB5R3 in bar graphs and note extremely low *P*-values supporting their conclusions, but it is unclear how these ratios are calculated. In most cases, the fraction of CYB5R3 that's modified is well below 10% (difficult to accurately assess because the unmodified form is so overexposed) yet their quantification shows ratios that must be normalized in some way that is not explained and not consistent. For example compare quantification of UFM1 modified CYB5R3 in Fig 2 with quantification of GFP-UFM1 modified CYB5R3 in Fig 5. Every figure in the paper has at least one panel using a similar quantification approach where the quantification does not seem to correspond to the actual data in the western blots.

Reply 5

We apologize for not specifying how we calculated the ratio of ufmylated CYB5R3 to CYB5R3. The signals of ufmylated CYB5R3 and CYB5R3 bands were quantified using Multi Gauge V3.2 (FUJIFILM Corporation), and the ratio of ufmylated CYB5R3 to CYB5R3 was calculated. The ratio was 1 when neither overexpression nor knockdown of any E3 components was performed (Fig. 2a and 2b). *P* values were determined by t-test (Fig. 2a) or Šidák's test after a two-way factorial analysis of variance (ANOVA) (Fig. 2b). The details are described in the corresponding Figure legends of the revised manuscript. In addition, according to a comment (comment 3) by Reviewer#5, in the y-axis of the graph, "UFM1~CYB5R3/CYB5R3" was substituted with "Relative value of the UFM1~CYB5R3/CYB5R3 ratio."

As the reviewer indicated, the amount of ufmylated CYB5R3 was low compared to that of free CYB5R3 under normal culture conditions. We need to find conditions for inducing CYB5R3 ufmylation such that anisomycin treatment (*i.e.*, protein synthesis inhibition) results in

135 ufmylation of RPL26 (Walczak et al., PNAS, 116:1299–1308, 2019; Wang et al., Cell Res, 30:5–20, 2019). However, we believe that this is beyond the scope of this study.

Currently we are not sure why CYB5R3 ufmylation is induced more effectively when using GFP-tagged UFM1 than FLAG-tagged or endogenous UFM1. However, our data show that the effect of the E3 component on CYB5R3 ufmylation is consistent regardless of Tag type.

140

Comment 6

In Fig 4 the authors use a pH-quenched fluorescent reporter attempt to demonstrate a role for UFMylation in promoting lysosomal degradation of CYB5R3. The visual data presented in panels b and c are impossible to interpret by eye so the authors quantify red puncta and use a statistical approach to claim a significant increase upon overexpression of UFL1/UFBP1 (and of course, the substrate cherry/GFP- CYB5R3) in starved cells. It is unclear why starvation was needed. Visual inspection of the graphical data show large error bars and many outlier points that are difficult to reconcile with the reported P-values, but I am not a statistician.

145

150 Reply 6

Most of the ER-phagy pathways identified thus far are induced by nutrient starvation and endoplasmic reticulum stress (Chino et al., Mol Cell, 74:909–921, 2019; Khaminets et al., Nature, 522:354–358, 2015; Stephani et al., eLife, 9:e58396, 2020), implying that ER-phagy induction requires a priming effect that activates autophagy-related proteins. Similarly, ER-phagy by UFM1 also requires this priming effect (*i.e.*, nutrient starvation). We discuss these points in the revised manuscript.

155

To quantify the number of RFP-positive punctae (the number of lysosomes encapsulating CYB5R3-positive ER) per cell, we used a Benchtop High-Content Analysis System (CQ1; Yokogawa Electric, Tokyo, Japan) and CellPathfinder software (Yokogawa Electric). This system allowed us to quantify fluorescently labeled lysosomes in multiple samples at the same time and without bias. The statistical analysis was conducted by Šidák's test after two-way ANOVA.

160

Comment 7

The authors have -somewhat arbitrarily decided to consider the third component of the tripartite UFM1 E3, CDK5RPAP3, as a “lysosome autophagy adaptor” based on data in a single report, while ignoring a substantial body of data supporting the view that this protein is a core constituent of the E3. Indeed, functional insight into the role of this E3 component was

165

170 recently published on BioRxiv. While it can't be ruled-out that CDK5RAP3 could have two
functions, it is concerning that its function in UFM1 ligase activity was not considered in the
experimental design or in the data interpretation. It is difficult to understand their rationale
for performing all the reconstitution and overexpression experiment with only two of the three
components of this E3 (and indeed they show, disconcertingly, that the E3 is entirely
175 dispensable for CYB5R3 UFMylation *in vitro*). In Fig 5a they finally test the role for
CDK5RAP3 and find that its inclusion in the overexpression paradigm actually reduces the
amount of UFMylated CYB5R3 (and UFBP1). While one explanation for this result is that
CDK5RAP3 promotes the specificity of the E3 for its authentic substrate (in part by suppressing
modification of non-authentic substrates) they choose to interpret the CDK5RAP3 effect as
supporting a role in lysosomal degradation. But the experiment they present in support of this
180 hypothesis, in Fig 5b (anti CYB5R3 blot) shows a remarkably weak effect of the lysosomal
inhibitor BafA (compare lanes 5 and 6). Although at best there is a 20% increase in the GFP-
UFM1- CYB5R3 band in the blot, strangely the quantification reports a highly significant ($P =$
0.003) effect.

185 Reply 7

First, we want to emphasize that we did not ignore BioRxiv
(<https://doi.org/10.1101/2022.01.31.478489>) or a paper that was recently published in *EMBO J*
(<https://pubmed.ncbi.nlm.nih.gov/36121123/>). Our manuscript was submitted in January,
whereas the paper in BioRxiv was submitted later.

190 As mentioned above, we were able to show both *in vivo* and *in vitro* that at least UFL1
and UFBP1 are required for ufmylation of CYB5R3 (please see Replies 1 and 2).

As the reviewer pointed out, CDK5RAP3 may act positively on authentic substrates
such as RPL26, and negatively on other substrates (Peter et al., *EMBO J* e111015). This is
supported by the fact that different proteins show either decreased or increased ufmylation in
195 *CDK5RAP3* knockout mouse tissues (Yang et al., *Development*, 146, dev169235, 2019). We
have described this possibility in the revised manuscript. Actually, we noticed that while
CDK5RAP3 had the ability to promote ufmylation of RPL26, it had an inhibitory effect on the
ufmylation of UFBP1 and CYB5R3 (data not shown; if the reviewer would like us to present
the data, we will do so as "Figure for Reviewer 1").

200 Statistical analysis of western blots was performed as described in Reply 5. The
signals of ufmylated CYB5R3 and CYB5R3 bands were quantified using Multi Gauge V3.2

(FUJIFILM Corporation), and the ratio of ufmylated CYB5R3 to CYB5R3 was calculated. P values were determined by Šidák's test after two-way ANOVA.

205 *Comment 8*

The structure/model in Fig 2g claims to show a steric clash that is not at all clear from the figure.

Reply 8

210 Based on the HS-AFM data, we manually modeled the open form of CYB5R3 and showed it side-by-side with the closed form (crystal structure) of CYB5R3 in the revised Fig. 2g. Lys214 (colored red) is located close to the FAD domain in the closed form and becomes more accessible to solvent in the open form. These observations suggest that ufmylation at Lys214 will bias the equilibrium between open and closed forms toward the open one. We added this discussion in the
215 revised manuscript.

Comment 9

*The claim that CYB5R3 reductase activity is lost from Fig 2i is not convincing. How are the results normalized? Do they have equivalent amounts of protein? What if the mutation doesn't
220 bind FAD as tightly?*

Reply 9

In Figure 2i, the same amount of CYB5R3 and ufmylated CYB5R3 proteins were used. Figure 2j shows that wild-type CYB5R3 (not ufmylated) and mutant CYB5R3 (K214R) have similar
225 reductase activity, implying that the K214 mutant has the same ability as wild type to possess FAD.

Reviewer #3:

230 General comment

All comments and concerns raised by the reviewer have been adequately addressed by the authors. I recommend publication of the revised manuscript.

Reply

235 Thank you very much for your positive evaluation.

Reviewer #4:

General comment

240 *There is no doubt that the manuscript is greatly improved and that the majority of concerns of the original reviewers have been addressed but a few points should be considered when making the final decision:*

Reply

245 Thank you very much for your positive evaluation and valuable suggestions that helped us improve the revised manuscript.

Comment 1

Reviewer 1's comments are largely addressed adequately bar the following minor point:

250 *- I think confirmation that the new endogenous CYB5R3 from microsomal fractions and probing for UFL1 is done under denaturing conditions would help. The methods suggest this is the case but I would make it more explicit in the text and figure legend. If it is not, it should be.*

Reply 1

255 We performed immunoprecipitation with the microsomal fraction denatured with 1% sodium dodecyl sulfate (SDS). Briefly, after cytoplasmic and microsomal fractionation, 10 µl of 10% SDS was added to 100 µl of both fractions. In the next step, 900 µl of IP buffer (20 mM Tris-HCl [pH 7.5], 150 mM NaCl, 1 mM EDTA, 1% NP40 and 1% TX-100), 2 µl of CYB5R3 antibody (GTX84646; GeneTex, Irvine, CA, USA), and 10 µl of Protein G Sepharose 4 Fast
260 Flow (Cytiva, Marlborough, MA, USA) were added to the 100 µl of cytoplasmic and microsomal fractions, and the mixture was mixed under constant rotation at 4°C for 3 h. The immunoprecipitates were washed five times with ice-cold IP buffer. The complex was boiled for 5 min in SDS sample buffer in the presence of β-mercaptoethanol to elute proteins. We describe the details in the Material and Methods section of the revised manuscript. We also indicated in
265 the corresponding figure legends that these experiments were performed under denatured conditions.

Comment 2

Reviewer 2's comments are largely addressed adequately bar the following minor points:

270 - there are still limitations on data showing the endogenous nature of some of the molecular
details of the mechanism. Immunofluorescence of UFL1 localisation remains absent, but on
balance it may be impossible to achieve this with current reagents. Short of knock-in of a tag to
the endogenous locus using CRISPR/Cas9, which would be extremely time consuming and has
no guarantee of methodological success, I think there is little else on balance that can be done
275 here. Overall, although this point has not been addressed completely, I don't think it should
preclude publication at this stage and in this journal.

Reply 2

Thank you for this comment. Unfortunately, our UFL1 antibody was unable to demonstrate
280 subcellular localization of endogenous UFL1. We agree that Tag knock-in to the UFL1 locus
would be a powerful tool, but we have not routinely performed such experiments in our lab and
they would be time consuming. Therefore, we decided to use the data from GFP-tagged UFL1.

Comment 3

285 - similarly, there is still a reliance on ectopic expression when using the Fluoppi system to show
ATG8 protein interaction with CDK5RAP3. Here I think the authors should at least provide
additional data to show that the expression level of these proteins is comparable with
endogenous and that mutation of known binding interfaces on ATG8 proteins, e.g. the LDS (if
these are indeed involved in CDK5RAP3 interaction) can reduce the Fluoppi signal, thus
290 validating the specificity of this system for detecting ATG8-adaptor/receptor interactions.

Reply 3

Expression levels of Azami Green-tagged CDK5RAP3 and its mutants (W269A, W294A,
W312A) were comparable to the endogenous one (Supplementary Fig. S8b in the revised
295 manuscript). Likewise, expression levels of Ash-tagged LC3B and GABARAP were also
similar to the endogenous ones (Supplementary Fig. S8b in the revised manuscript). In addition,
mutants of three LC3-interacting regions of CDK5RAP3 (W269A, W294A, W312A), which
have been reported to be required for interaction with LC3B or GABARAP (Stephani et al.,
eLife, 9:e58396, 2020), lost this interaction in the Fluoppi assay (Supplementary Fig. S8c in the
300 revised manuscript).

Comment 4

My additional comments - while I do not wish to add undue burden to the revision of this manuscript there is a fundamental point that is not specifically identified by any of the three prior reviewers but that I think is critical to draw the major conclusions of the manuscript. That is, proof of CYB5R3 turnover by autophagy is not per se proof of ER-phagy, despite the outer ER localisation of at least a substantial population of this protein. The authors should really use an abundant general ER marker and show degradation by ER-phagy occurs in an at least partially CYB5R3-dependent manner. For instance, ss-mRFP-GFP-KDEL, or tandem fluorescent RAMP4, etc., could be employed in a tandem fluorescent flux assay, analogously to how CYB5R3 is currently used. Similarly, the CYB5R3 positive autophagic foci identified by colocalisation with autophagy proteins should be demonstrated to contain other ER material via microscopy. N.B. I apologise if some of this was present in the previous version of the manuscript and has been removed, as I have not seen this draft. The response to Reviewer 1 might hint at this with regard to the mRFP-GFP-KDEL. I wasn't completely clear on this.

Reply 4

In the original manuscript, we utilized a doxycycline-inducible ER-phagy reporter containing an N-terminal ER signal sequence followed by tandem monomeric RFP and GFP sequences and the ER retention sequence KDEL (Chino et al., Mol Cell, 74:909–921, 2019). We found that UFL1 and UFBP1 expression significantly increased the number of RFP-positive punctae in both nutrient-rich and nutrient-deprived conditions. However, in the first round review of this manuscript, Reviewer#1 pointed out that the experiment with RFP-GFP-KDEL does not provide direct information about the role of CYB5R3, so we have used RFP-GFP-CYB5R3 and its mutants instead. We hope the reviewer understands this process.

Reviewer #5:

General comment

The revised manuscript has been improved significantly. However, as the reviewer #1 stated: the principal conclusion of the work – that CYB5R3 is a bona fide endogenous substrate of UFMylation is still unconvincing. He also raised many other issues, which are reasonable. Although almost all the experiments are done in the overexpressed system and the final in vivo experiments could not guarantee that the effect is caused by the ufmylation BUT NOT by other lysine modifications at the position 214, currently there are no better experiment techniques which could solve this problem.

There are still some minor issues in the manuscript, which need to be fixed before the manuscript is publishable. Because some issues require further experiments, therefore, I suggested a major revision of the manuscript. Please see below for other issues.

340

Reply

Thank you for your valuable comments that have helped us strengthen our revised manuscript. According to comments raised by Reviewers 1 and 5, we made the following changes.

345

1. To demonstrate the requirement for E3 in CYB5R3 ufmylation, we performed an *in vitro* ufmylation assay (Fig. 1g).
2. To prove that CYB5R3 is a *bona fide* substrate for UFM1, we performed several experiments to detect endogenous ufmylated CYB5R3 and knockdown-induced suppression of UFL1 or UFBP1 (Fig. 2a and b).
3. We delineated the quantification and statistical methods in this study.

350

For specific details, please see our reply to each comment below.

Comment 1

- *Figure 1a: The BiP distribution is different in the USFP2 KO cells? What is the reason for this? It should be discussed in the manuscript.*
- 355 • *Figure 1e: Why does the lower position have double bands? Figure 1f: Should CYB5R3-GFP-UFM1 be written as GFP-UFM1-CYB5R3?*
- *Figure 1i: Should CYB5R3-Strep-UFM1 be written as Strep-UFM1-CYB5R3?*

Reply 1

360

- In the fractionation confirmation in Fig. 1a, we believe that the Bip antibody we used in the blots was problematic. In fact, blots with anti-calnexin antibody (Santa Cruz Biotechnology, sc-46669), which has been used in many papers (*e.g.*, Rajagopalan et al., *Science*, 263:387-90, 1994), showed clearer fractionation regardless of the presence or absence of UFSP2.

365

- The double bands detected below ufmylated CYB5R3 in the blot using the UFM1 antibody in Fig. 1e are likely endogenous proteins (probably RPL26) to which UFM1 is covalently bound. In Fig. 1e of the revised manuscript, they are referred to as UFM1 conjugates.
- In accordance with the reviewer's comment, we have replaced CYB5R3~GFP-UFM1 with GFP-UFM1~CYB5R3 (Fig. 1h in the revised manuscript), and have substituted

370 CYB5R3~Strep-UFM1 with Strep-UFM1~CYB5R3 (Supplementary Fig. S6 in the revised
manuscript).

Comment 2

375 Figure 1d showed that UFL1 is important for the ufmylation of CYB5R3. However, in Figure 1f,
ufmylation of CYB5R3 occurs in the absence of UFL1. How could this happen? What caused
this discrepancy?

Reply 2

380 This *in vitro* reaction is a very specific experimental system, *i.e.*, an artificial system for producing
large amounts of ufmylated CYB5R3. First, it used an excess quantity of recombinant UFM1 (9
 μM), UBA5 (2 μM), and UFC1 (5 μM). Second, instead of purified CYB5R3 $\Delta\text{N}26$, a lysate of *E.*
coli expressing GST-CYB5R3 $\Delta\text{N}26$, was mixed with these recombinant enzymes. Under these
conditions, CYB5R3 was covalently bound to UFM1 without E3. We cannot explain the exact
mechanism involved, so we omitted this data from Figure 1. However, CYB5R3 covalently bound
385 by UFM1 is a valuable tool, so we used it for the pull-down experiments (Fig. 3b and g).

In the original manuscript, as pointed out by Reviewers 1 and 5, we agree that there
were problems in demonstrating the specificity of E3. To solve this issue, we expressed FLAG-
UFL1 and UFBP1-MYC in *UFC1*-deficient HEK293T cells and purified the FLAG-UFL1 and
UFBP1-MYC complex by immunoprecipitation with anti-FLAG antibody (anti-DDDDK-tagged
390 pAb-agarose). *In vitro* ufmylation assays were performed using recombinant UFM1, UBA5,
UFC1, the purified E3 complex, and a microsomal fraction prepared from *UFBP1*-deficient
HEK293T cells or recombinant CYB5R3 $\Delta\text{N}26$ from *E. coli*. As shown in Fig. 1g of the revised
manuscript, ufmylated CYB5R3 was not formed by the addition of UFM1, UBA5, and UFC1 to
microsomal fractions, but it was formed by adding them together with purified E3 complex. By
395 contrast, no ufmylated CYB5R3 was formed in the assay with recombinant CYB5R3 $\Delta\text{N}26$
(Supplementary Fig. S3c). This implies that E3 and one or more factors in the microsomal
fractions are required for ufmylation of CYB5R3. We presented the aforementioned points in the
revised manuscript.

400 Comment 3

- Figure 2a: Why is there no heavy chain but with intense light chain?
- Figure 2a and 2b: Add "Relative" in the y axis of the statistical data.

- 405
- *Figure 2b: IP and WB are required to confirm the ufmylation of endogenous CYB5R3. It is hard to tell whether the bands are specific or not if no IP is performed to eliminate the effect from other interacting proteins. The current images showed that there is a slightly less intense band when UFL1 and UFBP1 is knocked down in the anti-CYB5R3 blot. However, in the anti-UFM1 WB, this band is completely disappeared. This inconsistent might indicate this band is not ufmylated CYB5R3. A stricter experiment should be provided to confirm this result.*
- 410
- *Figure 2c: Why is the position for CYB5R3-HF in the microsome and cytosol different in the top image of Figure 2c?*
 - *Figure 2j: Why does not the KR mutation affect Vi in the absence of UFM1, UFL1, and UFBP1? This result indicates the effect is not physiologically relevant.*

415 Reply 3

Thank you for the valuable comments that helped us improve and strengthen our manuscript. As pointed out by Reviewers 1 and 5, the experiments presented in Fig. 2 include those that form the foundation of this study, and we substantially optimized these experiments with the latest attention.

- 420
- Mouse monoclonal antibody against CYB5R3 (GTX84646; GeneTex) was used for immunoprecipitation of endogenous CYB5R3 in microsomal fractions of HEK293T cells. The immunoprecipitants were subjected to immunoblot analysis with a rabbit monoclonal antibody against UFM1 (ab109305; Abcam). The secondary antibody used for its detection was Peroxidase AffiniPure Goat Anti-Rabbit IgG (H+L) (Jackson ImmunoResearch
- 425
- Laboratories, Code: 111-035-144). Therefore, we did not expect to detect either heavy or light chains. However, the product information for the secondary antibody states that it may cross-react with immunoglobulins of other animal species. Perhaps there is a cross-over to light chains in mice.
- 430
- According to the reviewer's suggestion, we added "Relative" to the y-axis of the statistical data in Fig. 2a and 2b.
- 435
- As shown in Fig. 2a of the revised manuscript, immunoprecipitation of microsomal fractions with anti-CYB5R3 antibody revealed endogenous ufmylated CYB5R3, and the same migrated band was significantly increased by overexpression of E3 (UFL1 and UFBP1). Furthermore, as shown in Fig. 2b of the revised manuscript, we showed that knockdown of *UFL1* or *UFBP1* significantly decreased the level of endogenous ufmylated

CYB5R3. These results were confirmed by immunoblotting with both anti-CYB5R3 and anti-UFM1 antibodies.

- CYB5R3 is anchored to the membrane by myristoylation of the second glycine at the N-terminus, and the 24 amino acids up to the N-terminal form the membrane-binding domain (Murakami et al., J Biochem, 105:312–7, 1989). Thus, CYB5R3 that was detected in the cytoplasmic fraction upon overexpression is considered to be the N-terminal truncated form. In fact, the CYB5R3 detected in this fraction is almost as mobile as $\Delta N26$, in which the N-terminal 26 amino acids are deleted. The aforementioned points are explained in the revised manuscript.
- Figure 2j shows that wild-type CYB5R3 (not ufmylated) and its mutant (K214R) have similar reductase activity, while ufmylated CYB5R3 has lost this activity.

Comment 4

- *Figure 3b and 3c: Why are there three bands in the top panel of Figure 3b?*
- *Blot or coomassie stain for GST should be provided in Figure 3b and 3c.*
- *Figure 3k: Which image, middle top image, was used for the quantification? Please indicate in the figure legend.*
- *Figure 3g: How can the authors rule out the interaction between UFM1 and UFBP1 when the authors demonstrate that the ufmylated CYB5R3 interacts with UFBP1? Based on the left panel of Figure 3g, UFBP1 interacts with UFM1.*

Reply 4

- As shown in Fig. 1i (Supplementary Fig. S6 in the revised manuscript), ufmylated CYB5R3 $\Delta N26$ is detected as a single band immediately after purification. The ufmylated CYB5R3 $\Delta N26$ shown in Fig. 1i (Supplementary Fig. S6 in the revised manuscript) was stored at -80°C for 5 months and utilized for the experiment in Fig. 3b. This means that the faster migrating ufmylated CYB5R3 $\Delta N26$ proteins detected in Fig. 3b are most likely degradation products that resulted from long-term storage and freeze–thawing. We have indicated that they are degradation products in the corresponding Figure legend of the revised manuscript. It would have been possible to produce and purify ufmylated CYB5R3 $\Delta N26$ again *in vitro* and to use the degradation product–free ufmylated CYB5R3 $\Delta N26$ in the experiments, but we decided not to do so because it would have required too much time and labor.

- 470
- In Fig. 3b and Fig. 3c, we prepared panels showing purified GST and GST-tagged proteins with Ponceau S staining.
 - The graph in Fig. 3k shows quantification of the blot with Crude presented in the upper left. This is noted in the corresponding Figure legend.
 - We do not intend to exclude the possibility that UFBP1 binds to free UFM1. In fact, as the reviewer pointed out, UFBP1 bound not only to ufmylated CYB5R3 but also to free UFM1 (Fig. 3g). We noted in the revised manuscript that UFBP1 has the ability to bind to free UFM1.
- 475

Comment 5

Figure 5e: Please include three different experimental conditions in the last bar graph.

480

Reply 5

According to this suggestion, data for three genotype mice are shown in Fig. 5e in the revised manuscript. One *Cyb5r3*^{+/+}, three *Cyb5r3*^{K214R/+}, and four *Cyb5r3*^{K214R/K214R} mice were used in this study. Since we used the data (thickness of cortex / distance from aqueduct to lateral edge) of the left and right cortices of the *Cyb5r3*^{K214R/+} and *Cyb5r3*^{K214R/K214R} mice and the right cortices of the wild-type and *Cyb5r3*^{K214R/K214R} mice, statistical analysis was performed on the wild-type and *Cyb5r3*^{K214R/+} mice together as controls, and these were compared to the *Cyb5r3*^{K214R/K214R} mice.

485

490 *Comment 6*

Supplementary Figure S5C: What is the lowest band in Western blotting image?

Reply 6

It was thought that it might be a degradation product of purified MBP-tagged UFL1. Therefore, we purified MBP-UFL1 again and performed the same experiment with the MBP-UFL1, and obtained reasonable results (disappearance of fast migrating bands).

495

Comment 7

Supplementary Figure S6: The fractionation is not perfect. GAPDH is also present in the microsomal fraction and is not consistent under different conditions. (It is not a big problem).

500

Reply 7

We acknowledge that the fractionation in Supplementary Fig. S6 (Supplementary Fig. S9 in the revised manuscript) is not perfect. Except for Supplementary Fig. S6 (Supplementary Fig. S9 in the revised manuscript), the fractionation experiments shown in this study were performed with HEK293T cells. By contrast, in Supplementary Fig. S6 (Supplementary Fig. S9 in the revised manuscript), we used mouse embryonic fibroblasts (MEFs) isolated from wild-type and *CYB5R3^{K214R/K214R}* knock-in mice. The detection of ufmylated CYB5R3 in MEFs was more difficult than in HEK293T cells, probably due to its low abundance. Therefore, detection required fractionating a large number of MEFs. As a result, there was some influx of the cytoplasmic fraction. As the reviewer pointed out, since this issue (some influx of the cytoplasmic fraction) is not essential and would require a great deal of effort to prevent, we did not redo this experiment during this revision. We hope that you understand.

515 *Comment 8*

Supplementary Table 1: Please explain the terms in detail. What are the peptides (95%)? Are they unique peptides or peptide spectrum matches?

Reply 8

520 “Peptides (95%)” indicates the number of distinct peptides that were identified with at least 95% confidence by ProteinPilot. We added this explanation as well as those for other columns in the text below Supplementary Table S1 and in Supplementary Dataset1, sheet “Index.”

Comment 9

525 *Supplementary Table 2 and Supplementary Table 3: Please pay attention to the significant digits.*

Reply 9

530 Thank you so much for pointing that out. We checked the significant digits of the raw data and corrected the tables accordingly.

Comment 10

Page 4, line 120: Why is UFL1 not required for CYB5R3 ufmylation?

535 Reply 10

On page 4, line 120 in the original manuscript we wrote the following: “When the E3 component UFL1 was co-overexpressed, we observed a band representing the MYC-UFM1~CYB5R3-His-FLAG conjugate, which was further enhanced by the expression of UFBP1 (Fig. 1D).” We concluded that UFL1 and UFBP1 are required for the ufmylation of CYB5R3 (Figs. 1g, 2a and 2b, etc., in the revised manuscript).

Comment 11

In the Discussion, the authors conclude that 2) ufmylated CYB5R3 interacts with the E3 ligase component UFBP1 to facilitate further ufmylation of CYB5R3. However, there is no sufficient evidence to conclude this statement.

Reply 11

We acknowledge that our model is not adequately substantiated by the data presented, as pointed out by the reviewer. In the revised manuscript, we have weakened our argument and removed this model from the main Figure (Fig. 5f) and moved it to Supplementary Fig. S10.

Comment 12

There are some typos for example, GFP-tagged UFL1 (GFP-UFL1) were...; bonafide

Reply 12

Thank you for the comment. We corrected the typos throughout the revised manuscript.

Reviewer #6:

General comment

My review of the manuscript is only based on the high-speed AFM part, given that this is my area of expertise. I will refrain from commenting on the much broader other aspect of the paper. Appreciate the efforts that the authors have put in to investigating CYB5R3 with high-speed AFM to observe structural changes. Unfortunately very little information is given in the manuscript about the HS-AFM data, which makes it very difficult to interpret. Based on what criteria did the authors attribute the closed and open states of the molecule? What kind of differences do the authors expect based on the crystallography data? It seems to me that the fluctuations comparing images denoted as "O" with each-other are similar to the fluctuations comparing "O" with "C", or "C" with "C". HS-AFM experiments always have significant differences from experiment to

570 *experiment. How many different experiments have the authors performed? How many molecules*
have they observed per experiment? In how many cases did they see a convincing difference
between "O" and "C"?

Reply

575 We would like to thank this reviewer for this important comment on our HS-AFM experiment.

> *Based on what criteria did the authors attribute the closed and open states of the molecule?*

We divided the images into two groups (the closed and open states) depending on the distances
between the two lobes of the molecule (FAD and NADH domains). In the initial version of the
580 manuscript, we used a tip-scan-type HS-AFM apparatus with 1.0-nm/pixel resolution. To obtain
clearer images, we re-examined CYB5R3 with a sample-scan-type HS-AFM apparatus. In the
revised manuscript, we obtained CYB5R3 images with 0.6-nm/pixel resolution. Because the
molecules sometimes change their orientation on the mica surface, it was difficult to divide all
the images into two groups (the closed and open states). Therefore, among all the images obtained,
585 we chose and analyzed those in which the two lobes (FAD and NADH domains) were facing up.
We measured the distances between the two lobes and obtained histogram data using Kodec
software. All of the histograms from six molecules were well reproduced by double-Gaussian
distribution whose peaks were at 2.62 ± 0.07 and 4.71 ± 0.11 nm. On the other hand, we simulated
AFM images of CYB5R3 (PDB ID: 1umk) using BioAFMviewer software. Then we obtained the
590 cross-sectional profile of the two lobes in the simulated AFM image. The distance between the
two lobes was ~2.5 nm, a value close to one of the two peaks of the fitted curve of the real HS-
AFM data. In the revised manuscript, representative images of the two conformations are shown
in Fig. 2f with simulated AFM data. We manually modeled an open conformation of CYB5R3 by
changing the arrangement between the NADH and FAD domains at the hinge region (Leu147) in
595 order to yield a predicted AFM image similar to the experimental one (Fig. 2f and g).

> *How many different experiments have the authors performed? How many molecules have they*
observed per experiment?

600 In the revised manuscript, we performed three different experiments and collected data from six
different molecules in total.

> *In how many cases did they see a convincing difference between "O" and "C"?*

605 Based on the histograms of the distances between two lobes, we observed the open state and the closed state at similar frequencies.

REVIEWERS' COMMENTS

Reviewer #4 (Remarks to the Author):

I believe that my concerns have been addressed.

I would recommend that in the final revision that the data with the standard ER-phagy reporter (ss-mRFP-GFP-KDEL) that was removed from the first draft of the manuscript is now presented.

Reviewer #5 (Remarks to the Author):

The revised manuscript is significantly improved. Below are some minor issues that the authors should take care of.

1. It is hard to tell that the regulation of ER-phagy is indeed caused by the ufmylation of CYB5R3. Similarly, it is not convincing that the microcephaly occurred in the Cyb5r3K214R/K214R knock-in mice is caused by the deficiency of its ufmylation. One cannot rule out the effect of other types of post-translational modifications on lysine residues. Therefore, it is highly suggested that the authors should also discuss the limitations of their study in the discussion section.
2. Figure 4b and c: Please add magnification or enlarge for the middle panels.
3. Figure 5c: Please add information for the color code at the left side of the images.

Reviewer #6 (Remarks to the Author):

The authors have adequately addressed my concerns. The new images are much more convincing.

Reviewer #4:

General comment

I believe that my concerns have been addressed.

5 Reply

Thank you for your detailed and thorough peer review.

Comment 1

10 *I would recommend that in the final revision that the data with the standard ER-phagy reporter (ss-mRFP-GFP-KDEL) that was removed from the first draft of the manuscript is now presented.*

Reply 1

15 Thank you for this suggestion. During the revision, we performed the ER-phagy assay with ss-mRFP-GFP-KDEL using a Benchtop High-Content Analysis System and CellPathfinder software without bias. Although ER-phagy tended to be induced by ufmylation of CYB5R3, we did not recognize significant difference in the assay with mRFP-GFP-KDEL (Supplementary Figure S8 in the revised manuscript). We concluded that since ufmylated CYB5R3 is involved in ER-phagy for restricted ER subpopulation, ssRFP-GFP-KDEL that locates in whole ER is
20 unable to monitor the ufmylated CYB5R3-mediated ER-phagy. We stated this in the result section in the revised manuscript.

Reviewer #5 (Remarks to the Author):

General comment

25 *The revised manuscript is significantly improved.*

Reply

Thank you very much for your positive evaluation.

30 Minor comments

Comment 1

*It is hard to tell that the regulation of ER-phagy is indeed caused by the ufmylation of CYB5R3. Similarly, it is not convincing that the microcephaly occurred in the Cyb5r3K214R/K214R knock-in mice is caused by the deficiency of its ufmylation. One cannot rule out the effect of
35 other types of post-translational modifications on lysine residues. Therefore, it is highly*

suggested that the authors should also discuss the limitations of their study in the discussion section.

Reply 1

40 As the reviewer pointed out, there is a possibility that at least ufmylation-defective *Cyb5r3* knock-in mice cause microcephaly due to effects other than ufmylation. Therefore, in the discussion section of the revised manuscript, we explained as follows:

Though we do not exclude a possibility that the mutation inhibits other post-translational modification(s) except for the ufmylation, our results suggest that the defect in macro ER-phagy
45 through the ufmylation of CYB5R3 is involved in the pathogenesis of RCM type II.

Comment 2

Figure 4b and c: Please add magnification or enlarge for the middle panels.

50 Reply 2

Thank you for the comment. According to this suggestion, we add “Enlarge” in the middle panels (Figure 5b and c in the revised manuscript).

Comment 3

55 *Figure 5c: Please add information for the color code at the left side of the images.*

Reply 3

We added the information at the left side of the images (Fig. 5b, c e and Fig. 6c in the revised manuscript).